# Preserving Expert-Level Privacy in Offline Reinforcement Learning

**Navodita Sharma**[*]                                            *navoditasharma@google.com*
*Google DeepMind*

**Vishnu Vinod**[*]                                               *vishnuvinod2001@gmail.com*
*CeRAI, IIT Madras*

**Abhradeep Thakurta**                                           *athakurta@google.com*
*Google DeepMind*

**Alekh Agarwal**                                                *alekhagarwal@google.com*
*Google Research*

**Borja Balle**                                                  *bballe@google.com*
*Google DeepMind*

**Christoph Dann**                                               *chrisdann@google.com*
*Google Research*

**Aravindan Raghuveer**                                          *araghuveer@google.com*
*Google DeepMind*

**Reviewed on OpenReview:** *https://openreview.net/forum?id=2bjOeVgCdO*

## Abstract

The offline reinforcement learning (RL) problem aims to learn an optimal policy from historical data collected by one or more behavioural policies (experts) by interacting with an environment. However, the individual experts may be privacy-sensitive in that the learnt policy may retain information about their precise choices. In some domains like personalized retrieval, advertising and healthcare, the expert choices are considered sensitive data. To provably protect the privacy of such experts, we propose a novel consensus-based expert-level differentially private offline RL training approach compatible with any existing offline RL algorithm. We prove rigorous differential privacy guarantees, while maintaining strong empirical performance. Unlike existing work in differentially private RL, we supplement the theory with proof-of-concept experiments on classic RL environments featuring large continuous state spaces, demonstrating substantial improvements over a natural baseline across multiple tasks.

## 1 INTRODUCTION

Leveraging existing offline datasets to learn high-quality decision policies via offline Reinforcement Learning (RL) is a critical requirement in many settings ranging from robotics (Fu et al., 2020; Levine et al., 2020) and recommendation systems (Bottou et al., 2013; Ie et al., 2019; Cai et al., 2017) to healthcare applications (Oberst & Sontag, 2019; Tang et al., 2022). Correspondingly, there is now a rich literature on effective techniques (Fujimoto et al., 2019b; Kumar et al., 2019; 2020; Cheng et al., 2022) for learning from such offline datasets, collected using one or more *behavioural policies*. An often overlooked aspect of this literature is,

---

[*]These authors contributed equally.

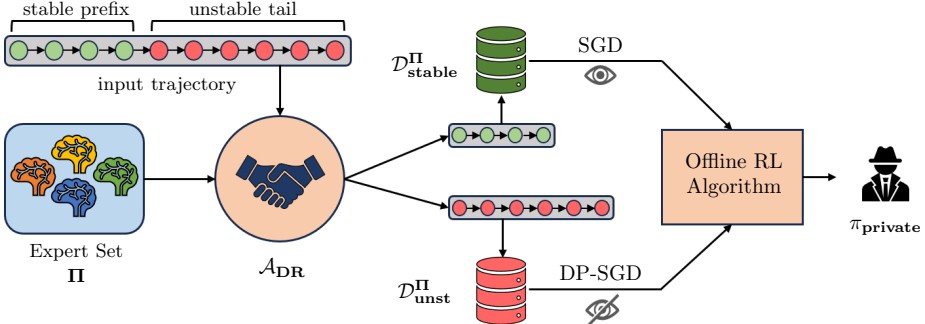

Figure 1: Training pipeline for Expert-level Differentially Private Offline RL given an expert set $\Pi = \{\pi_1, \pi_2, \ldots, \pi_m\}$ and input trajectories logged into an offline dataset $D_\Pi$. $\mathcal{A}_{DR}$ (Algorithm 2) splits input trajectories and adds stable prefixes to $D^\Pi_{stable}$ discarding the rest to $D^\Pi_{unst}$. These are used to train any off-the-shelf offline RL algorithm (see Algorithm 3) and learn an expert-level differentially policy $\pi_{private}$.

however, that in privacy sensitive domains such as personalized retrieval, advertising, and healthcare, the behavioural policies might reveal private information about the preferences or strategies used by the corresponding experts (users, advertisers, health care providers etc.) whose decisions underlie the offline data. In such scenarios, we ideally seek to uncover the broadly beneficial strategies employed in making decisions by a large cohort of experts whose demonstrations the offline data is collected with, while not leaking the private information of any given expert. We call this novel setting *offline RL with expert-level privacy*, and design algorithms with strong privacy guarantees for it.

There has been a growing interest in differentially private RL, both in online (Wang & Hegde, 2019; Qiao & Wang, 2023a; Zhou, 2022; Liao et al., 2023) and offline (Qiao & Wang, 2023b) settings. The offline RL literature, which is most relevant to this work, primarily focuses on protecting privacy at the *trajectory-level*. However, when each expert contributes multiple trajectories to the demonstration dataset, private information can still be leaked under trajectory-level differential privacy. Protecting the privacy of an expert in this scenario requires significantly more careful techniques. Secondly, the prior works mostly focus on tabular or linear settings (Qiao & Wang, 2023a;b; Zhou, 2022), where the offline RL problem can be effectively solved using count or linear regression based techniques, and is hence amenable to existing techniques for privatizing counts or linear regression models. In contrast, we make no assumptions on the size of the state space, the parameterization of the learned policy or the policy underlying expert demonstrations. For a more detailed discussion of related work, we refer the reader to Section 2.

**Contributions:** With this context, our paper makes the following contributions:

- We identify and formalize the problem of offline RL with expert-level privacy and motivate it with practical examples. We also explain the inadequacy of prior algorithms in this setting.

- We provide *practical algorithms* for offline RL with strong expert-level privacy guarantees. In contrast with prior approaches which use noisy statistics of the data in offline RL (Qiao & Wang, 2023b) or rely on very strict assumptions like tabular settings or linear function approximators (Wang & Hegde, 2019), limiting real-world applicability, we remove these constraints, supporting any gradient-based offline RL algorithm with general function approximators and continuous state spaces.

- DP-SGD (Abadi et al., 2016) is the go-to method today for privately training gradient-based general function approximators. We adapt DP-SGD for expert-level privacy as a natural baseline in our problem setting. Our algorithm identifies a subset of the data which can be used in learning without any added noise, and significantly improves upon the DP-SGD baseline since it *selectively adds noise to gradients only where it is needed*, unlike DP-SGD which uniformly noises all gradient updates. We evaluate our algorithm on standard RL benchmarks, and discuss the results in Section 6.

**Our Algorithm:** Our algorithm proceeds in two stages; see Figure 1. The first is a data filtering stage that produces a set of trajectory prefixes (denoted $D_{stable}^{\Pi}$), using a variant of the Sparse Vector Technique (SVT) for privacy accounting (Dwork & Roth, 2014), that can be used in training without any further noise addition, in a privacy-preserving manner. The intuition is that members of this set are likely to occur across many experts, and hence using it in training does not violate the privacy of an individual expert. The second stage involves selectively running DP-Stochastic Gradient Descent (DP-SGD)[1] (Song et al., 2013; Bassily et al., 2014; Abadi et al., 2016), adapted for expert-level privacy on the remainder of the trajectories, $D^{\Pi} \setminus D_{stable}^{\Pi}$. Our empirical results strongly show that combining both stages of the algorithm described above, performs better than either of them individually. The only assumption we make on the underlying offline RL algorithm is that it is gradient-based, so that we can use DP-SGD to privatize its updates.

**Motivation:** We motivate our problem setting using the case of financial portfolio management. Different fund managers employ proprietary and privacy-sensitive investment strategies that guide their trading decisions. In such a setting, treating the algorithms used by individual fund managers as expert policies and learning an expert-level private investment strategy (policy) using offline RL will be beneficial as a starting strategy for new investors and other parties that do not have specialized investment strategies.

Alternatively, in the case of political campaign optimization in a multi-party democracy, large political parties may use proprietary algorithms to maximize their reach, whereas independent candidates, often lacking the same resources, are placed at a disadvantage. Access to a privately learnt aggregate campaigning strategy will allow such players to compete on a somewhat equal footing, allowing voters greater effective choice.

At a high-level, our setting is similar to the 'warm-start' problem in optimization (Sambharya et al., 2024), translated to an offline RL setting where data is privacy-sensitive and cannot be directly trained on.

**Note on Experts:** Throughout the paper, we use the term "expert" to refer to RL agents, policies, or algorithms used in various real-world settings. They do not refer to *human experts* like doctors, trading agents etc. We learn an expert-level DP policy, from the offline trajectories logged by such agents interacting with the environment like behavioural policies in offline RL (Levine et al., 2020; Prudencio et al., 2023).

## 2   RELATED WORK

**Comparison to Prior Work on Private Offline RL:** There have been recent works exploring the intersection of differential privacy (DP) and reinforcement learning (RL) (Wang & Hegde, 2019; Qiao & Wang, 2023a;b; Zhou, 2022; Liao et al., 2023). Qiao & Wang (2023a;b); Zhou (2022); Wang & Hegde (2019) explore the problem either in the tabular setting or in the linear function approximation setting. In either of the cases there is a sufficient statistic (like discrete state-action visit counts or feature covariances to privatize sensitive data) which if privatized, would suffice to privatize the whole algorithm. Specifically, Wang & Hegde (2019) explores the use of differential privacy (DP) in continuous spaces to only protect the value function approximator. Liao et al. (2023) explores RL with linear function approximation under local differential privacy guarantee, a stronger notion of privacy where users contributing data do not trust the central aggregator. Meanwhile, Qiao & Wang (2023a;b); Vietri et al. (2020) work with tabular MDP settings and discrete state-action spaces, often utilizing some form of state-action visit counts to privatize sensitive data. In our work, we extend existing approaches beyond the state-space and linear function approximator constraints and propose a general algorithm which can be used with *continuous state spaces*, and with general functional approximation for RL tasks.

To the best of our knowledge, expert-level DP has not been dealt with in the context of RL so far. Prior works often limit themselves to exploring trajectory-level privacy, which has finer granularity of privacy protection which makes the setting arguably "easier". In this work, we adopt an expert-level notion of neighbourhood when a large number of behavioural policies (or experts) are used for data collection. We further note that expert-level privacy differs from existing privacy notions like Joint DP (Chowdhury & Zhou, 2022), or other notions which aim to protect the transitions and rewards of the MDP.

---

[1]DP-SGD is to an SGD based approach where gradients coming from individual experts are clipped, and appropriate Gaussian noise is added to ensure DP.

**Background on DP-SGD and Sparse-Vector Technique:** DP-SGD first introduced in Song et al. (2013); Bassily et al. (2014), and adapted for deep learning by Abadi et al. (2016), proposed gradient noise injection to guarantee differentially private model training. Several variants of DP-SGD have been proposed such as McMahan et al. (2017), which allows user-level DP training in federated learning scenarios and Kairouz et al. (2021) a private variant of the Follow-The-Regularized-Leader (DP-FTRL) algorithm that compares favourably with amplified DP-SGD (Abadi et al., 2016; Balle et al., 2018).

The Sparse-Vector Technique (Roth & Roughgarden, 2010; Hardt & Rothblum, 2010; Dwork & Roth, 2014), developed over a sequence of works, allows one to carefully track the privacy budget across queries, and provide tighter data-dependent privacy guarantees as opposed to vanilla composition (Dwork & Roth, 2014). At a high-level, the idea is to identify "stable queries", which have low local sensitivity and do not change their answers when moving to a neighboring dataset. Such queries can be answered without paying any additional privacy cost. In Algorithm 2, we use a modification of this idea to output parts of the expert trajectories that are "stable" without incurring any additional privacy cost (in $\varepsilon$).

**Relationship with Offline RL Literature:** Our approach is agnostic to the underlying offline RL algorithm used and can be combined with any off-the-shelf model-based (Kidambi et al., 2020; Yu et al., 2020; 2021) or model-free (Kumar et al., 2019; 2020; Cheng et al., 2022) offline RL algorithms which use gradient updates to learn an optimal policy. We focus on model-free approaches in our experiments.

## 3 PRELIMINARIES

### 3.1 Offline RL

Offline RL (Levine et al., 2020; Prudencio et al., 2023) is a data-driven approach to RL that aims to circumvent the prohibitive cost of interactive data collection in many real-world scenarios. Under this paradigm, a static dataset collected by some (possibly sub-optimal) behaviour policy $\pi_\beta$ is used to learn a policy $\pi$ without any interaction with the environment.

Let the environment be modeled by a Markov Decision Process (MDP) $\mathcal{M} = \langle S, A, r, P, \rho, \gamma \rangle$, where $S$ is the state space, $A$ is the action space, $r : S \times A \to \mathbb{R}$ is the reward function, $P : S \times A \to \Delta(S)$ is the transition kernel ($\Delta(\mathcal{X})$ is the set of all probability distributions over $\mathcal{X}$), $\rho \in \Delta(S)$ is the initial state distribution and $\gamma \in [0, 1]$ is the discount factor.

**Definition 3.1** (Offline RL). Given an environment represented by the MDP $\mathcal{M} = \langle S, A, r, P, \rho, \gamma \rangle$ defined above and a dataset $D = \{(s_i, a_i, r_i, s'_i)_i\}_{i=1}^n$ generated with $(s_i, a_i) \sim \mu$ for some state-action distribution $\mu$, $r_i = r(s_i, a_i)$ and $s'_i \sim P(\cdot|s_i, a_i)$, the objective is to find an optimal policy $\pi^* : S \to \Delta(A)$, defined as

$$\pi^* = \arg\max_\pi \mathbb{E}^\pi_{s_1 \sim \rho}[R_\tau]$$

, where $R_\tau = \sum_{t=1}^{|\tau|} \gamma^t r_t$ is the discounted return of trajectory $\tau$, generated from $\pi$ and $\mathcal{M}$ as: $s_1 \sim \rho$, $a_t \sim \pi(.|s_t)$, $r_t = r(s_t, a_t)$ and $s_{t+1} \sim P(.|s_t, a_t)$ for $t \geq 1$ upto $|\tau|$ timesteps.

A key concern in offline RL is that the state-action distribution encountered in the offline data to learn a policy $\pi$ might significantly differ from that encountered upon actually executing $\pi$ in the MDP. There is a host of existing offline RL methods that address this challenge in different ways (Kumar et al., 2019; Fujimoto et al., 2019b; Kumar et al., 2020; Cheng et al., 2022). We use BCQ (Fujimoto et al., 2019b) and CQL (Kumar et al., 2020) as our offline RL methods due to their strong performance across a variety of tasks in prior evaluations and popularity in existing offline RL literature..

We consider a scenario where the dataset contains demonstrations collected from a number of different behavioural policies or experts with varying degrees of optimality. We denote the set of behaviour policies by $\Pi = \{\pi_1, \ldots, \pi_m\}$. Imposing the constrain of expert-level privacy can conflict with the rewards attainable by the learned policy. For example, if most experts are similar but a single, high-reward expert diverges from them, an inherent tradeoff arises between expert-level privacy and utility. We only consider the privatization of the expert policies (the chosen actions). Thus, information about the environmnent dynamics such as the transition kernel and reward function are not protected under our proposed algorithm.

## 3.2 Differential Privacy

Differential privacy (DP) (Dwork et al., 2006; Dwork & Roth, 2014) quantifies the loss of privacy associated with data release from any statistical database. Informally, a procedure satisfies DP if the distribution of its outputs on two datasets which differ only in one record is very similar. We use approximate-DP (or $(\varepsilon, \delta)$-DP) throughout our work. This notion is formally defined below.

**Definition 3.2** $((\varepsilon, \delta)$-DP). For positive real numbers $\varepsilon > 0$ and $0 \le \delta \le 1$, an algorithm $\mathcal{A}$ is $(\varepsilon, \delta)$-DP iff, for any two neighbouring datasets $D, D' \in \mathcal{D}^*$ ($\mathcal{D}$ is the space of all data records), and $S \subset Range(\mathcal{A})$:

$$Pr(\mathcal{A}(D) \in S) \le e^{\varepsilon} Pr(\mathcal{A}(D') \in S) + \delta \tag{1}$$

where $Range(\mathcal{A})$ is the set of all possible outputs of $\mathcal{A}$.

Definition 3.2 guarantees that the probability of seeing a specific output on any two neighbouring datasets can differ at most by a factor of $e^{\varepsilon}$, with a relaxation of $\delta$. In most ML applications reasonable privacy guarantees have $\varepsilon \le 10$ and $\delta \le \frac{1}{n}$ (Ponomareva et al., 2023), where $n$ is the cardinality of the dataset.

One can instantiate Definition 3.2 with different of neighbourhood relations. Definition 3.3 describes our setting of *expert-level privacy* and the more commonly studied *trajectory-level privacy*.

**Definition 3.3** (Expert- and Trajectory-level privacy). Let $\mathcal{D}$ be the domain of all valid trajectories possible in the underlying MDP. Let $D^{\Pi} \in \mathcal{D}^*$ be a data set of trajectories generated by a set of experts $\Pi$. $D^{\Pi}, D^{\Pi'} \in \mathcal{D}^*$ are *expert-level neighbours* if the set of experts $|\Pi \Delta \Pi'| = 1$, where $\Delta$ is the set difference. Alternatively, if $D, D' \in \mathcal{D}^*$ are sets of trajectories and $|D \Delta D'| = 1$, then $D$ and $D'$ are *trajectory-level neighbours*.

We note that the key difference between expert and trajectory level privacy notions, as defined above is that removing an expert $\pi$ can remove up to $|D^{\pi}|$ trajectories from the overall dataset, where $D^{\pi}$ is the set of trajectories contributed by $\pi$. This is a much larger change than allowed in trajectory-level privacy, and a naïve treatment would result in paying an additional cost due to group privacy (Dwork & Roth, 2014), scaling linearly in the number of trajectories each expert contributes. At the same time, as we argue below, via a set of settings, expert-level privacy is often a critically required notion of privacy protection.

## 3.3 Problem Setup

**Dataset.** Let $\Pi$ denote a set of $m$ privacy-sensitive policies, corresponding to $m$ experts, $\Pi = \{\pi_1, \ldots, \pi_m\}$. $D^{\Pi} = D^{\pi_1} \cup \cdots \cup D^{\pi_m}$ is the aggregated dataset generated by the interaction of the $m$ experts with the environment. Each $D^{\pi_i}$ is a set of $N$ trajectories $\tau = (s_1, a_2, s_2, a_2, \ldots, s_L, a_L, s_{L+1})$ of length $L$ obtained by logging the interactions of policy $\pi_i$ with the MDP $\mathcal{M} = \langle S, A, r, P, \rho_0, \gamma \rangle$, where $s_1 \sim \rho_0$, $a_h \sim \pi_i(\cdot|s_h)$ and $s_{h+1} \sim P(\cdot|s_h, a_h)$. We denote by $|\tau|$ the length of the trajectory as the number of full state-action pairs it consists of. Further, we use $\tau_{h:h'} = (s_h, a_h, \ldots s_{h'}, a_{h'}, s_{h'+1})$, where $h \le h'$, to denote the sub-trajectory of $\tau$ starting from the $h$-th timestep until the $h'$-th, including the trailing next state $s_{h'+1}$.

**Note on Trajectory Notation:** As in Definition 3.1, trajectories are classically defined as a series of $(s, a, r, s')$ tuples, where the next state $s'$ of one transition is the state $s$ of the next transition. Since the rewards associated with transitions are not used in the privacy analysis or in the filtering stage of our algorithm, we simplify the classical notation for our analysis and proofs as described above.

**Goal.** The goal is to learn a policy using an offline RL algorithm with expert-level DP guarantees on $D^{\Pi}$, thus ensuring that the policy learnt in this manner is not much different from a policy learnt using $D^{\Pi'}$.

We make the following mild assumptions in the design of our solution.

**Assumption 3.4.** The action space $A$ is discrete.

This is a fairly mild assumption satisfied in many offline RL settings. This arises while identifying stable trajectory prefixes for our algorithm, and we leave an extension to continuous action spaces to future work.

**Assumption 3.5.** We can query the sensitive expert policies, i.e., evaluate $\pi_i(a|s)$ for a given $i, s$ and $a$.

Similar assumptions are reasonable to make in setups involving learning a student model from an ensemble of privacy-sensitive teacher models, where access to the teacher models are assumed (e.g. Papernot et al. (2017)). The privacy-sensitive experts in our setting are analogous to these privacy-sensitive teacher models. Again, query access only comes up in computing the stability of the trajectory prefix, and can likely be relaxed to using behavior-cloned versions of the experts with a slightly worse utility bound.

**Assumption 3.6.** Let $\Pi_s$ be the class of expert policies considered by our algorithm. Hence, $\Pi \subseteq \Pi_s$. We assume a *minimum action probability* ($p_{min}$), defined as, $p_{\min} \leq \min_{\pi \in \Pi_s} \inf_{s \in S} \min_{a \in A} \pi(a|s)$

This value is used in the data filtering stage (first stage) of our algorithm. Note that we can always take $p_{min} = 0$. However, our algorithm is more meaningful when $p_{min} > 0$. Examples of common policy classes with $p_{min} > 0$ include $\varepsilon$-greedy policies and softmax policies with bounded parameters.

Existing work in DP-RL relies on strict assumptions like tabular settings or linear function approximation, limiting real-world applicability. Our approach removes these constraints and extends to any state-of-the-art gradient-based offline RL algorithm with general function approximators and continuous state spaces.

## 4  ALGORITHM

For a given trajectory prefix $\tau$ , we define the total probability of $\tau$ being generated by all the experts as: $count_\tau(\Pi) = \sum_{i=1}^{m} \left[ \prod_{j=1}^{L_\tau} \pi_i(a_j|s_j) \right]$ [2], where $L_\tau = |\tau|$, is the length of trajectory $\tau$. Algorithm 1 deems $\tau$ stable if $count_\tau(\pi)$ exceeds a carefully chosen threshold, implying that enough experts are likely to generate $\tau$. These stable prefixes may be used in training an offline RL algorithm without any noise addition. We need Assumption 3.4 for this step. Given a trajectory, the count function gives the sum of probabilities of each expert traversing it. We use $count_\tau(\Pi)$ to check if enough experts can generate this trajectory.

---

**Algorithm 1** PrefixQuery ($\mathcal{A}_{PQ}$): Tests if count of experts expected to execute a trajectory is large enough

---

**Require:** Expert policies $\Pi$, trajectory $\tau = (s_1, a_1, s_2, a_2, \dots)$, stability threshold $\hat{\theta}$, $\varepsilon$
 1: Compute expected experts: $count_\tau(\Pi) \leftarrow \sum_{\pi \in \Pi} \prod_{j=1}^{|\tau|} \pi(a_j|s_j)$
 2: **if** $count_\tau(\Pi) + Lap(\frac{4}{\varepsilon}) > \hat{\theta}$ **then return** $\top$
 3: **else return** $\bot$

---

Algorithm 2 uses Algorithm 1 as a subroutine, traversing up to $T$ trajectories to find stable prefixes. The dataset $D_{stable}^{\Pi}$ returned by Algorithm 2 is composed of these stable trajectory prefixes. The discarded tails of each trajectory (including several full trajectories that were discarded) are collected in another dataset:

$$D_{unst}^{\Pi} = \{\tau_{k+1:|\tau|} \text{ if } \tau_{1:k} \in D_{stable}^{\Pi} \text{ for } k \geq 1 \text{ or } \tau, \ \ \tau : D^{\Pi}\}$$

Consider any parameterized gradient-based offline RL algorithm $\mathcal{H}_\phi$, where $\phi$ denotes the parameters. Training this algorithm consists of using $(s, a, r, s')$ tuples collected in the offline dataset to update the parameters (by splitting the trajectories into into individual transitions). Having split the dataset into stable and unstable trajectories, we can now leverage the stability of $D_{stable}^{\Pi}$ to accumulate the gradient updates of $\mathcal{H}_\phi$ on transitions drawn from this dataset without any noise addition. The full algorithm, detailing the training using a mix of noisy gradient updates for transitions from $D_{unst}^{\Pi}$ and noiseless gradient updates for transitions from $D_{stable}^{\Pi}$ is given in Algorithm 3.

The sampling probability $p$ controls the expected fraction of batches sampled from the unstable dataset $D_{unst}^{\Pi}$. The DP-SGD updates of the form DP-SGD($\phi_n, \mathcal{B}, \varepsilon_2, \delta_2, N$) in Algorithm 3 refer to noisy gradient updates with additive noise scaled as to ensure ($\varepsilon_2, \delta_2$) expert-level DP. To do so we add the modifications to the standard DP-SGD algorithm (Abadi et al., 2016):

- During gradient updates, we ensure that an expert contributes *at most one transition* to a batch.

---

[2]The full probability of being generated by the given set of experts, is actually proportional to the count function multiplied by the state-transition probabilities (which do not change for neighbouring experts).

---

**Algorithm 2** DataRelease ($\mathcal{A}_{DR}$): Releasing public dataset after privatisation

---

**Require:** Expert dataset $D^\Pi$, expert policies $\Pi$, $\varepsilon_1$, $\delta_1$, min action-selection probability $p_{min}$, unstable
   query cutoff $T \leq N \cdot m$
1: $L \leftarrow \max\limits_{\tau \in D^\Pi} |\tau|$
2: $\varepsilon' \leftarrow \frac{\varepsilon_1}{\sqrt{32T \log(2/\delta_1)}}, \quad \delta' \leftarrow \frac{\delta_1}{(2TL)}$
3: $c_{min} \leftarrow \frac{e^{\varepsilon'}}{(e^{\varepsilon'}-1)}, \quad \theta \leftarrow \frac{c_{min}}{p_{min}}, \quad D^\Pi_{stable} \leftarrow \{\}$
4: Reshuffle $D^\Pi$
5: **for** $c = 1, \ldots, T$ **do**
6: $\quad \tau \leftarrow$ next trajectory from $D^\Pi$
7: $\quad \hat{\theta} \leftarrow \theta + \frac{4}{\varepsilon'} \log(1/\delta') + Lap(\frac{2}{\varepsilon'})$
8: $\quad$ **for** $i = 1$ to $L$ **do**
9: $\quad\quad r \leftarrow \mathcal{A}_{PQ}(\Pi, \tau_{1:i}, \hat{\theta}, \varepsilon')$
10: $\quad\quad$ **if** $r = \top$ and $i = |\tau|$ **then** $D^\Pi_{stable} \leftarrow D^\Pi_{stable} \cup \{\tau_{1:i}\}$
11: $\quad\quad$ **else if** $r = \bot$ **then**
12: $\quad\quad\quad$ **if** $i > 1$ **then** $D^\Pi_{stable} \leftarrow D^\Pi_{stable} \cup \{\tau_{1:i-1}\}$
13: $\quad\quad\quad$ **break**
14: **return** $D^\Pi_{stable}$

---

- To account for privacy amplification while sampling batches from the dataset, we use the use the fraction of experts from which we sample. Hence, the subsampling coefficient $q$ for DP-SGD analysis is $q = p \cdot \frac{|B|}{m}$, where $m$ is the number of experts.

We detail the modified expert-level DP-SGD, used as a baseline in Section 6, in Appendix B. See Section 5 for the privacy analysis for the overall pipeline combining Algorithms 2 and 3.

---

**Algorithm 3** Selective DP-SGD for Offline RL

---

**Require:** Stable dataset $D^\Pi_{stable}$, Unstable dataset $D^\Pi_{unst}$, Sampling probability $p$, Gradient-based offline
   RL algorithm $\mathcal{H}_\phi$, $\varepsilon_2, \delta_2$, $N$ training steps, $B$ batch size
1: Initialize $\phi_0$
2: **for** $n = 1$ to $N$ **do**
3: $\quad b \sim \text{Bernoulli}(p)$
4: $\quad$ **if** $b = 1$ **then** $\phi_{n+1} \leftarrow \text{DP-SGD}(\phi_n, \mathcal{B}, \varepsilon_2, \delta_2, N)$, where we sample batch $\mathcal{B} \sim D^\Pi_{unst}$
5: $\quad$ **else if** $b = 0$ **then** $\phi_{n+1} \leftarrow \text{SGD}(\phi_n, \mathcal{B})$, where we sample batch $\mathcal{B} \sim D^\Pi_{stable}$
6: **return** $\mathcal{H}_{\phi_N}$

---

**Relationship with User-level DP:** The relationship between trajectory-level and expert-level DP is analogous to the notions of example-level and user-level privacy in existing DP literature (Lévy et al., 2021; Zhou et al., 2022; Zhao et al., 2024). However, we retain naming conventions from existing RL literature.

We note that the expert-level DP-SGD (discussed above and in Appendix B) update step in Line 6 of Algorithm 3 is compatible with other user-level DP algorithms that can be used for training on the unstable data instead. Most practical approaches for achieving user-level DP are based on clipping user-level gradients to bound user contributions (Lévy et al., 2021; Bassily & Sun, 2023; Liu & Asi, 2024; De et al., 2022). These approaches have been empirically tested to be successful on large scale datasets (Charles et al., 2024). Other approaches for achieving user-level DP are mostly theoretical in nature and come with restrictions (such as on the function class) that limit their practical usage. The baseline used for comparison in our paper (described in Appendix B) is a representative of this class of approaches, and a natural way to tackle our problem statement in such a way that we generalize to any function approximators and continuous, high-dimensional state spaces. We demonstrate that our method outperforms this baseline by identifying a subset of the data where noisy training is not required, before running the same baseline on the rest of the data.

**Sub-Optimality Guarantees for the Learned Policy:** While we show in the following section that the Algorithm 3 provides expert-level privacy, we do not provide explicit bounds on the suboptimality of the returned policy. Typical offline RL literature shows that using pessimistic approaches, the learned policy is competitive with any other policy whose state-action distribution is well-covered by the offline data distribution (Xie et al., 2021; Cheng et al., 2022). However, our training distribution undergoes two changes from the *a priori* offline data distribution. First is that the stable prefixes might be distributionally rather different from the entire state-action distribution, and potentially emphasize the states and actions near the start of a trajectory, as shorter prefixes are more likely to be stable. Second, the gradient clipping in DP-SGD adds an additional bias. Understanding the effect of latter with arbitrary function approximation is beyond the scope of this paper. We do, however, study the effects of distributional biases from adding varying amounts of the stable prefixes in our experiments, and find that they are typically very beneficial, particularly in tasks where the optimal behaviour does not change significantly over time.

## 5 PRIVACY ANALYSIS

We note that all the trajectories may be of various lengths. However, as we shall see in the analysis, the privacy cost in the $\delta$ parameter scales linearly with the length of the trajectory, i.e., $\delta_1 \propto L\delta'$ (Line 2 in Algorithm 2), where $\delta'$ is the per-timestep cost and $\delta_1$ is the full cost of the trajectory. By setting $L$ to be the maximum length across all trajectories, we carry out a worst-case analysis of the privacy leakage. For a fixed per-timestep privacy budget, we overestimate the total privacy cost by safely considering the worst-case (maximum) trajectory lengths. In other words, for the benefit of the analysis, we assume that all trajectories have the same (maximal) length $L = \max_{\tau \in D^\Pi} |\tau|$.

Algorithm 2 can be thought of as $T$ iterations of a combination of two algorithms $\mathcal{A}_1$ and $\mathcal{A}_2$:

- $\mathcal{A}_1$ samples a random expert $\pi \sim \mathrm{Unif}(\Pi)$ and a trajectory $\tau \sim D^\pi$ and converts it into $|\tau|$ queries, $Q^\tau = \{count_{\tau_1}(.), \ldots, count_{\tau_{|\tau|}}(.)\}$, where $\tau_i = \tau_{1:i}$ is an $i$-length prefix of the trajectory $\tau$.

- $\mathcal{A}_2$ is invoked on $\Pi$ and the query set $Q^\tau$, running an iteration of the Sparse-Vector Technique (Dwork & Roth, 2014) (SVT). $\mathcal{A}_2$ treats $\Pi$ as the privacy-sensitive database, $Q^\tau$ as the query sequence, and $\hat{\theta}$ as the threshold for SVT. $\mathcal{A}_2(\Pi, Q^\tau)$ outputs $r \in \{\top, \bot\}^{k+1}$ such that $r[i] = \top \ \forall \ i \leq k$ and $r[k+1] = \bot$, where $k \leq L$ is the length of the stable prefix. Note that $r[i] = \mathcal{A}_{PQ}(\Pi, \tau_{1:i}, \hat{\theta}, \varepsilon')$.

- $\mathcal{A}_1$, upon receiving $r$, releases the prefix $\tau_{1:k} = (s_1, a_1, \ldots, s_k, a_k, s_{k+1})$.

*For a fixed stream of queries $Q^\tau$*, Algorithm $\mathcal{A}_2$ is $(\varepsilon', 0)$-DP since it just invokes one round of the Sparse Vector Algorithm (Dwork & Roth (2014)). Note that the trajectories are just used for creating the queries for SVT, and the set of expert policies $\Pi$ is the privacy database. For a given trajectory $\tau$, $\mathcal{A}_2$ identifies the longest prefix which is stable, or in other words, has enough experts which are likely to sample this prefix. Note that, the privacy cost $\varepsilon'$ for identifying this prefix is independent of the length of the trajectory.

However, $Q^\tau$ itself is a function of the set of experts $\Pi$. Despite this, in the following analysis, we show that $\mathcal{A} = \mathcal{A}_1 \circ \mathcal{A}_2$ is also differentially private (due to the carefully chosen threshold in Algorithm 2 Line 6). We then compose the privacy guarantees over $T$ trajectories, since $\mathcal{A}_{DR}$ is simply $\mathcal{A}$ applied $T$ times.

We now state two key lemmas for proving that $\mathcal{A}$ is differentially private. For a given trajectory prefix $\tau_k$ of length $k$, let $E_\tau$ denote the event that $\mathcal{A}_1$ samples any trajectory $\tau'$ containing $\tau_k$ as a prefix, i.e., $\tau'_{1:k} = \tau_k$.

**Lemma 5.1.** *For a trajectory prefix $\tau_k$, and neighbouring expert sets $\Pi, \Pi'$, if $count_{\tau_k}(\Pi) \geq c_{min}$ then,*

$$\Pr(E_{\tau_k}|\Pi) \leq e^{\varepsilon'} \Pr(E_{\tau_k}|\Pi') \tag{2}$$

*where $c_{min}$ and $\varepsilon'$ are as defined in Algorithm 2.*

If the probability of seeing a trajectory prefix $\tau_k$ across all experts is large enough, then the probability that $\mathcal{A}_1$ samples any trajectory with prefix $\tau_k$ is stable when one expert changes. See Appendix A for proof.

**Lemma 5.2.** *For any trajectory prefix $\tau_k$, and expert set $\Pi$ such that $count_{\tau_k}(\Pi) < \theta$ the following holds:*

$$\Pr(\mathcal{A}_{PQ}(\Pi, \tau_k, \hat{\theta}, \varepsilon') = \top) < \delta' \tag{3}$$

*where $\varepsilon', \delta', \theta, \hat{\theta}$ are as defined in Algorithm 2.*

Informally, if the total probability of seeing a trajectory prefix $\tau_k$ across all experts is *not* large enough, then the probability that it is labelled as stable by Algorithm 1 is less than $\delta'$. See Appendix A for proof.

**Theorem 5.3.** *Under Assumption 3.4, $\mathcal{A}$ is $(2\varepsilon', \frac{\delta_1}{2T})$-DP, where $\varepsilon', \delta_1$ are defined in Algorithm 2.*

*Proof Sketch:* The main result used to prove this theorem is the following:

$$\Pr(\mathcal{A}(\Pi) = \tau) \le e^{2\varepsilon'} \Pr(\mathcal{A}(\Pi') = \tau) + \Pr(E_\tau|\Pi) \cdot \delta' \tag{4}$$

which is obtained from Lemma 5.4 that we give below.

Let $\mathcal{T}$ denote the set of all possible trajectory prefixes using $S, A$. To prove that $\mathcal{A}$ is $(2\varepsilon', \delta_1/(2T))$-differentially private, we need to prove the following for any pair of neighbouring $\Pi, \Pi' \subseteq \Pi_{p_{min}}$:

$$\sum_{\tau \in \mathcal{T}} \max\{\Pr(\mathcal{A}(\Pi) = \tau) - e^{2\varepsilon'} \Pr(\mathcal{A}(\Pi') = \tau), 0\} \le \frac{\delta_1}{2T}.$$

The above can be proved using equation 4, and we defer the full proof to Appendix A.

The following lemma gives the key result used to prove Theorem 5.3.

**Lemma 5.4.** *For a trajectory prefix $\tau_k$ and neighboring expert sets $\Pi, \Pi'$, if $count_{\tau_k}(\Pi) < c_{min}$ then,*

$$\Pr(\mathcal{A}(\Pi) = \tau_k) < \Pr(E_{\tau_k}|\Pi) \cdot \delta'. \tag{5}$$

*If $count_{\tau_k}(\Pi) \ge c_{min}$ then,*

$$\Pr(\mathcal{A}(\Pi) = \tau_k) \le e^{2\varepsilon'} \Pr(\mathcal{A}(\Pi') = \tau_k) + \Pr(E_{\tau_k}|\Pi) \cdot \delta' \tag{6}$$

*Proof Sketch:* The proof of this lemma follows from Lemmas 5.1 and 5.2. Let $r = \top \cdots \top \bot \in \{\top, \bot\}^{k+1}$ and $\gamma^{(L)} = \top^L$. When $k < L$ we have

$$\Pr(\mathcal{A}(\Pi) = \tau_k) = \sum_{a \in A} [\Pr(\mathcal{A}_2(\Pi, Q^{\tau_k \cdot a}) = r) \Pr(E_{\tau_k \cdot a}|\Pi)]$$

where $\tau_k \cdot a$ denotes the trajectory prefix $\tau_k$ followed by action $a$. And when $k = L$,

$$\Pr(\mathcal{A}(\Pi) = \tau_k) = \Pr(\mathcal{A}_2(\Pi, Q_k^\tau) = \gamma^{(L)}) \Pr(E_{\tau_k}|\Pi)$$

Consider the case $k = L$. If $count_{\tau_k}(\Pi)$ is small enough, then $\Pr(\mathcal{A}_2(\Pi, Q_k^\tau) = \gamma^{(L)})$ is less than $\delta'$ (Lemma 5.2) and we get equation 5. On the other hand, if $count_{\tau_k}(\Pi)$ is greater than $c_{min}$, then $\Pr(E_{\tau_k}|\Pi)$ is bounded by Lemma 5.1. Using that $\mathcal{A}_2$ is always $\varepsilon'$-DP when the queries are fixed we get equation 6. The case of $k < L$ is less straightforward, but can also be analyzed by considering two cases for each $a \in A$, $count_{\tau_k \cdot a}(\Pi) \ge c_{min}$ or $count_{\tau_k \cdot a}(\Pi) < c_{min}$. See Appendix A for the full proof. $\square$

**Theorem 5.5.** *Under Assumption 3.4, Algorithm 2 is $(\varepsilon_1, \delta_1)$-DP.*

*Proof.* Algorithm 2 executes $\mathcal{A}$ with $(\varepsilon_1/\sqrt{8T \log(2/\delta_1)}, \delta_1/(2T))$-DP (see Theorem 5.3), $T$ times. Using advanced composition (Corollary 3.21 in Dwork & Roth (2014)) we get that Algorithm 2 is $(\varepsilon_1, \delta_1)$-DP. $\square$

Using all the above, we derive the following expert-level privacy guarantee for the overall training pipeline.

**Theorem 5.6.** *Algorithm* 3 *is* $(\varepsilon, \delta)$-*DP, where:*

$$\varepsilon = \varepsilon_1 + \varepsilon_2 \quad and \quad \delta = \delta_1 + \delta_2 \tag{7}$$

*Proof.* Follows from sequential composition of DP guarantees for the release of $D_{stable}^{\Pi}$ and DP-SGD training on samples from $D_{unst}^{\Pi}$. $\square$

**Note on Novelty:** We present a novel, useful and non-trivial extension of the Sparse Vector Technique (SVT) to preserve expert-level DP in Offline RL. The stream of queries in vanilla SVT, (Algorithm 1, Section 3.6, Dwork & Roth (2014)), is not dependent on the private database. However, in our setup, the stream of queries is a function of the set of the experts (which is the private database to be protected), and directly applying SVT does not ensure privacy. However, we show that despite the query stream being a function of the private database, SVT can work if the threshold (line 6 in Algorithm 2) is chosen in a certain way. We use the probability distribution of the query stream derived from the experts' policies to prove this.

## 6 EXPERIMENTS

Ensuring reasonable privacy requires significant hyperparameter tuning and a large number of data points, often impacting performance adversely[3]. Intuitively, with the demand for expert-level privacy in our setting, we risk severe degradation of utility since each "record" (each expert's data) forms a much larger fraction of the overall dataset. We recover most of this performance loss, significantly outperforming DP-SGD.

**Dataset Generation:** We train 3000 experts each on the Cartpole, Acrobot, LunarLander and HIV Treatment environments. Cartpole, Acrobot and LunarLander are from the Gymnasium package (Towers et al., 2023). The HIV Treatment simulator is based on the implementation by Geramifard et al. (2015) of the model described in Ernst et al. (2006). Experts are trained on variations of the default environment, created by modifying the values of key parameters (e.g. gravity in LunarLander environment). The trained experts are then used to generate demonstrations on a default setting of the environment, collectively forming the aggregated dataset for offline RL. This training methodology ensures that the experts show varied (often sub-optimal) behaviours on the default environment. As shown in Figure 2, the experts' return-histograms indicate that the mix of experts we create shows significant diversity in the policies learnt. Further details regarding the environment settings used to train the expert policies are available in Appendix B.



Figure 2: Expert return histograms, with Kernel Density Estimation, for experts trained on heterogeneous (left to right) LunarLander, Acrobot, CartPole and HIV Treatment environments. We use experts with wide ranges of performance on the test environment. Return values for HIV treatment normalised by $10^6$.

**Training Setup:** Our algorithm allows us to use any off-the-shelf gradient-based offline RL algorithm, and we demonstrate this by experimenting with Conservative Q-Learning (CQL) (Kumar et al., 2020) and Batch Constrained Deep Q-Learning (BCQ) (Fujimoto et al., 2019b). More specifically, we use the DQN (Deep Q-Network) version of CQL, and the discrete-action version of BCQ (Fujimoto et al., 2019a), since we operate under the assumption of discrete action spaces. To represent each of the Q-value functions in both the algorithms and the generative model in BCQ, we use a neural network with 2 hidden layers of 256 units each. During training, we perform a grid search to find the optimal set of hyper-parameters (learning rate, batch size, sampling probability $p$, DP-SGD noise) for each environment; see Appendix B.

---

[3]In supervised learning, for example, the accuracy on ImageNet training can go from $> 75\%$ to $\sim 47.9\%$ in the private case (Kurakin et al., 2022), under example-level privacy.

The maximum trajectory length is fixed to 200 for Acrobot and CartPole in the collected demonstrations. While evaluating the final learned policy, the maximum episode length is again set as 200 for Acrobot, we increase this to 1000 for Cartpole for increased difficulty in an otherwise "simple" environment. We allow all DP-SGD training runs to progress until the privacy budget is used up completely.

**Baselines:** We compare our method against DP-SGD adapted for expert-level privacy. A brief discussion of the changes we made are presented in Section 4. A full discussion, along with analysis, is presented in Appendix B. We keep the underlying RL algorithm same for fair comparison. In the absence of prior methods for the expert-level privacy setting, we limit our empirical studies to this naive baseline. We also train a non-private policy over the whole dataset using vanilla CQL/BCQ (ignoring privacy concerns). This serves as an empirical upper-bound on the performance achievable under privacy constraints. In our experiments, we report episodic returns, normalized to lie between 0 (random policy) and 1 (optimal policy) averaged over 10 evaluation runs at the end of training, as a fraction of the return of the non-private baseline.

## 6.1 Results

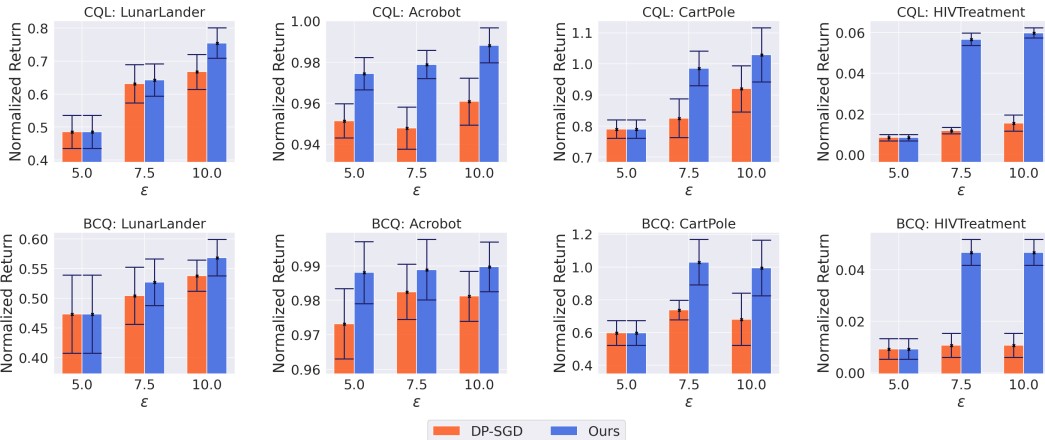

Figure 3: Performance of our method and DP-SGD for different values of $\varepsilon$ with $\delta = 1/m$, where $m$ is the number of experts. We report the episodic return, normalized between 0 (random policy) and 1 (optimal policy) averaged over 10 evaluation runs (with 95% confidence intervals) at the end of training, as a fraction of the non-private baseline. Our method consistently outperforms DP-SGD, especially in the high $\varepsilon$ regions.

We observe empirically that our algorithm, which leverages expert consensus to select stable trajectory prefixes prior to offline RL training, shows superior performance to merely using an adapted expert-level DP-SGD algorithm. Figure 3 compares the performance of our method with DP-SGD across different values of $\varepsilon$. All the results are averaged over 3 runs and 95% confidence intervals are also reported. Our method is able to recover much of the performance loss due to DP-SGD, especially in the high $\varepsilon$-regions.

We experiment with $\varepsilon < 5$ too, but for such low $\varepsilon$, Algorithm 2 yields very little stable data, which does not meaningfully influence training. Thus, the model primarily learns from the unstable data. Thus, setting $\varepsilon_1 = 0$ and $\varepsilon_2 = \varepsilon$ is optimal as we minimize the noise injected by the DP-SGD training. This is in contrast to the observations for higher $\varepsilon$ values where we take $\varepsilon_1 = \frac{3\varepsilon}{4}, \varepsilon_2 = \frac{\varepsilon}{4}$ for LunarLander and CartPole. For Acrobot and HIVTreatment, we set $\varepsilon_1 = \varepsilon, \varepsilon_2 = 0$, that is, we only train on $D_{stable}^\Pi$. Similarly, we split $\delta$ as $\delta_1 = \frac{9\delta}{10}, \delta_2 = \frac{\delta}{10}$. These results for low $\varepsilon$ are presented in Table 2.

We also note the optimal settings of sampling probabilities in Table 1. Interestingly, for Acrobot (with CQL and BCQ) and HIV Treatment (with CQL) environments, just $D_{stable}^\Pi$ suffices to get the optimal performance in high $\varepsilon$-region and we did not need to pay any privacy cost of DP-SGD during model training. We also report the variation of performance with various mixes of the two datasets (by varying $p$) in Table 2.

We also evaluate the effect of varying the number of experts on the performance of our algorithm relative to DP-SGD, in figure 4, on the Acrobot environment. As $m$ increases, the likelihood that $count_\tau(\Pi)$ exceeds

| Environment | CQL | | | BCQ | | |
|---|---|---|---|---|---|---|
| | $\varepsilon = 5$ | $\varepsilon = 7.5$ | $\varepsilon = 10$ | $\varepsilon = 5$ | $\varepsilon = 7.5$ | $\varepsilon = 10$ |
| **LunarLander** | 1.0 | 0.8 | 0.8 | 1.0 | 0.5 | 0.5 |
| **Acrobot** | 0.0 | 0.0 | 0.0 | 0.0 | 0.0 | 0.0 |
| **CartPole** | 1.0 | 0.9 | 0.8 | 1.0 | 0.5 | 0.5 |
| **HIV Treatment** | 1.0 | 0.0 | 0.0 | 1.0 | 0.9 | 0.9 |

Table 1: Best choices for probability of sampling from $\mathcal{D}^{\Pi}_{unstable}$ ie. $p$ for Algorithm 3 on test environments

| Environment | CQL | | | BCQ | | |
|---|---|---|---|---|---|---|
| | $p = 0.0$ | $p = 0.8$ | $p = 1.0$ | $p = 0.0$ | $p = 0.5$ | $p = 1.0$ |
| LunarLander | INF | 0.75 | 0.67 | INF | 0.57 | 0.54 |
| CartPole | 0.67 | 1.03 | 0.92 | 0.31 | 0.99 | 0.67 |

| Environment | $\varepsilon$ | $\varepsilon_1 = \frac{3}{4}\varepsilon$ | $\varepsilon_1 = 0$ |
|---|---|---|---|
| Acrobot (BCQ) | 2.5 | 0.74 | 0.95 |
| Cartpole (BCQ) | 2.5 | 0.28 | 0.58 |
| Cartpole (BCQ) | 5.0 | 0.56 | 0.60 |

Table 2: We perform ablation studies with low, intermediate and high sampling probabilities $p$ from the unstable dataset $D^{\Pi}_{unst}$, at $\varepsilon = 10$. INF (infeasible) denotes that the stable dataset $D^{\Pi}_{stable}$ was empty, and training was infeasible. We also vary the allocation of $\varepsilon$ in the low-privacy regime ($\varepsilon \leq 5$) and report utility.

the noisy threshold $\hat{\theta}$ also increases, and with it the number and length of trajectory prefixes in the stable dataset $D^{\Pi}_{stable}$. Thus, performance scales with the number of experts. Conversely, for very low values of $m$, the stable dataset is empty and our algorithm defaults towards the baseline.

**Note on Implementation:** In the implementation of both our method and DP-SGD, we create batches by by shuffling datasets of individual experts and randomly interleaving them such that each batch has at most one data point from each expert. This differs slightly from the poisson sampling based accounting in our privacy analysis. This gap between analysis and implementation in our work corresponds to the gap between privacy amplification by subsampling and shuffling. A promising line of work (Erlingsson et al., 2019; Feldman et al., 2021) aims to derive comparable theoretical amplification guarantees via shuffling. Empirically, however, the gap in utility is expected to be minimal (Ponomareva et al., 2023).

We use the PLD accountant from the `dp-accounting` library to account for the privacy loss in DP-SGD training.

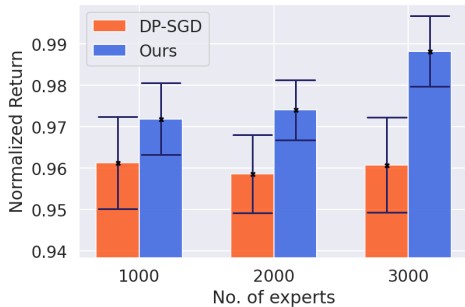

Figure 4: Performance of our method and DP-SGD for $m = 1000, 2000$ and 3000 experts, on the Acrobot environment with $\varepsilon = 10.0$ and CQL as the underlying offline RL algorithm. We observe increasing improvements of our method over DP-SGD as $m$ increases.

# 7 CONCLUSION

In this work, we initiate the study of offline RL with expert-level privacy. We describe the novel challenges of the setting, and provide theoretically sound algorithms which are evaluated in proof-of-concept evaluation. In future work, we would like to relax Assumptions 3.4-3.6, by replacing the queries to the experts with behaviour cloned versions and extending our trajectory prefix selection paradigm to continuous action spaces (for e.g using kernel density estimation). Understanding the utility implications of Algorithm 3, despite the influence of our privatization scheme on the training distribution is another important avenue for future work. Further, privacy auditing in the reinforcement learning setting is another important open research problem orthogonal to our work. Finally, while our empirical evaluation convincingly establishes the promise of this approach, we leave a larger evaluation across diverse and larger-scale RL benchmarks for future studies.

## Acknowledgements

The authors thank Krishna Pillutla and Balaraman Ravindran for fruitful discussions and helpful comments. We also thank the TMLR reviewers for their helpful feedback. This research was supported in part by the Post-Baccalaureate Fellowship at the Centre for Responsible AI, IIT Madras.

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

# A Theorem Proofs

## A.1 Proof of Lemma 5.1

**Lemma 5.1.** *For a trajectory prefix $\tau_k$, and neighbouring expert sets $\Pi, \Pi'$, if $count_{\tau_k}(\Pi) \geq c_{min}$ then,*

$$\Pr(E_{\tau_k}|\Pi) \leq e^{\varepsilon'} \Pr(E_{\tau_k}|\Pi') \tag{2}$$

*where $c_{min}$ and $\varepsilon'$ are as defined in Algorithm 2.*

*Proof.* Recall $E_{\tau_k}$ is the event a trajectory with prefix $\tau_k$ is generated. Assuming $\tau_k = s_1, a_1, s_2, \ldots, a_k, s_{k+1}$, the probability that a fixed expert $\pi$ generates such a prefix when interacting with the MDP is $\rho_0(s_1) \left( \Pi_{j=1}^k \pi(a_j|s_j) P(s_{j+1}|s_j, a_j) \right)$. Since the expert is chosen uniformly at random, we have:

$$\Pr(E_{\tau_k}|\Pi) = \frac{1}{m} \sum_{i=1}^m \left[ \rho_0(s_1) \left( \Pi_{j=1}^k \pi_i(a_j|s_j) P(s_{j+1}|s_j, a_j) \right) \right] \tag{8}$$

$$= \frac{1}{m} \rho_0(s_1) \left( \Pi_{j=1}^k P(s_{j+1}|s_j, a_j) \right) \sum_{i=1}^m \left( \Pi_{j=1}^k \pi_i(a_j|s_j) \right) \tag{9}$$

$$= \frac{1}{m} \rho_0(s_1) \left( \Pi_{j=1}^k P(s_{j+1}|s_j, a_j) \right) count_{\tau_k}(\Pi) \ . \tag{10}$$

Similarly,

$$\Pr(E_{\tau_k}|\Pi') = \frac{1}{m'} \rho_0(s_1) \left( \Pi_{j=1}^k P(s_{j+1}|s_j, a_j) \right) count_{\tau_k}(\Pi') \ . \tag{11}$$

Where $m' = |\Pi'|$. Since $|\Pi \Delta \Pi'| = 1$,

$$|count_{\tau_k}(\Pi) - count_{\tau_k}(\Pi')| \leq 1 \quad \text{and} \quad |m - m'| = 1 \ . \tag{12}$$

This gives us the following,

$$\frac{\Pr(E_{\tau_k}|\Pi)}{\Pr(E_{\tau_k}|\Pi')} = \frac{count_{\tau_k}(\Pi) \times m'}{count_{\tau_k}(\Pi') \times m} \tag{13}$$

$$\leq \max\left( \frac{count_{\tau_k}(\Pi)(m+1)}{(count_{\tau_k}(\Pi)+1)m}, \frac{count_{\tau_k}(\Pi)(m-1)}{(count_{\tau_k}(\Pi)-1)m}, \frac{m+1}{m}, \frac{m-1}{m} \right) \tag{14}$$

$$= \max\left( \frac{count_{\tau_k}(\Pi)(m-1)}{(count_{\tau_k}(\Pi)-1)m}, \frac{m+1}{m} \right) \ . \tag{15}$$

Note that, since $m + 1 > count_{\tau_k}(\Pi)$,

$$\frac{m+1}{m} < \frac{count_{\tau_k}(\Pi)}{count_{\tau_k}(\Pi)-1} \tag{16}$$

$$\max\left( \frac{count_{\tau_k}(\Pi)(m-1)}{(count_{\tau_k}(\Pi)-1)m}, \frac{m+1}{m} \right) < \frac{count_{\tau_k}(\Pi)}{count_{\tau_k}(\Pi)-1} \ . \tag{17}$$

This allows us to bound the ratio of probabilities of $E_{\tau_k}$ for $\Pi$ and $\Pi'$ as

$$\frac{\Pr(E_{\tau_k}|\Pi)}{\Pr(E_{\tau_k}|\Pi')} < \frac{count_{\tau_k}(\Pi)}{count_{\tau_k}(\Pi)-1} \leq e^{\varepsilon'} \ , \tag{18}$$

where the last inequality uses $count_\tau(\Pi) \geq c_{min} = e^{\varepsilon'}/(e^{\varepsilon'} - 1)$. $\qquad \square$

### A.2 Proof of Lemma 5.2

**Lemma 5.2.** *For any trajectory prefix $\tau_k$, and expert set $\Pi$ such that $count_{\tau_k}(\Pi) < \theta$ the following holds:*

$$\Pr(\mathcal{A}_{PQ}(\Pi, \tau_k, \hat{\theta}, \varepsilon') = \top) < \delta' \tag{3}$$

*where $\varepsilon', \delta', \theta, \hat{\theta}$ are as defined in Algorithm 2.*

*Proof.*

$$\Pr(\mathcal{A}_{PQ}(\Pi, \tau, \hat{\theta}, \varepsilon') = \top) = \Pr\left[count_{\tau_k}(\Pi) + Lap(4/\varepsilon') > \theta + (4/\varepsilon')\log(1/\delta') + Lap(2/\varepsilon')\right] \tag{19}$$
$$< \Pr\left[\theta + Lap(4/\varepsilon') > \theta + (4/\varepsilon')\log(1/\delta') + Lap(2/\varepsilon')\right] \tag{20}$$
$$= \Pr\left[Lap(4/\varepsilon') > (4/\varepsilon')\log(1/\delta') + Lap(2/\varepsilon')\right] \tag{21}$$

Consider $u \sim Lap(4/\varepsilon')$ and $v \sim Lap(2/\varepsilon')$. For a fixed $v$, we have,

$$\Pr\left[u \geq (4/\varepsilon')\log(1/\delta') + v\right] \leq \frac{\delta'}{2} e^{-v(\varepsilon'/4)} \tag{22}$$

Using the Laplace tail bound, we account for the randomness of $v$ from equation 21, equation 22 and the probability density function of the Laplace distribution, we have,

$$\Pr\left[Lap(4/\varepsilon') > (4/\varepsilon')\log(1/\delta') + Lap(2/\varepsilon')\right] \leq \int_{-\infty}^{\infty} \left(\frac{\delta'}{2} e^{-v(\varepsilon'/4)} \times \frac{\varepsilon'}{4} e^{-|v|(\varepsilon'/2)}\right) dv \tag{23}$$
$$< \delta' \tag{24}$$

This gives the statement of the lemma.

$\square$

### A.3 Proof of Lemma 5.4

**Lemma 5.4.** *For a trajectory prefix $\tau_k$ and neighboring expert sets $\Pi, \Pi'$, if $count_{\tau_k}(\Pi) < c_{min}$ then,*

$$\Pr(\mathcal{A}(\Pi) = \tau_k) < \Pr(E_{\tau_k}|\Pi) \cdot \delta'. \tag{5}$$

*If $count_{\tau_k}(\Pi) \geq c_{min}$ then,*

$$\Pr(\mathcal{A}(\Pi) = \tau_k) \leq e^{2\varepsilon'} \Pr(\mathcal{A}(\Pi') = \tau_k) + \Pr(E_{\tau_k}|\Pi) \cdot \delta' \tag{6}$$

*Proof.* We split this proof across 3 lemmas. Lemma A.1 proves this bound for the case $count_{\tau_k}(\Pi) < c_{min}$ and $k < L$. Lemma A.2 handles the case $count_{\tau_k}(\Pi) \geq c_{min}$ and $k < L$. Finally, Lemma A.3 proves this for $k = L$. Combining these 3 gives the statement of Lemma 5.4. $\square$

**Lemma A.1.** *For any trajectory prefix $\tau_k$, $k < L$, and neighboring expert sets $\Pi, \Pi' \subseteq \Pi_{p_{min}}$, if $count_{\tau_k}(\Pi) < c_{min}$ then,*

$$\Pr(\mathcal{A}(\Pi) = \tau_k) < \Pr(E_{\tau_k}|\Pi) \cdot \delta'$$

*where $\delta'$ is as defined in Algorithm 2.*

*Proof.* Let $r = \top \cdots \top \bot \in \{\top, \bot\}^{k+1}$

$$\Pr(\mathcal{A}(\Pi) = \tau_k) = \sum_{a \in A} \left[\Pr(\mathcal{A}_2(\Pi, Q^{\tau_k \cdot a}) = r) \Pr(E_{\tau_k \cdot a}|\Pi)\right] \tag{25}$$

where $\tau_k \cdot a$ denotes the trajectory prefix $\tau_k$ followed by action $a$.

$\mathcal{A}_2$ runs the Sparse Vector Algorithm using the the query sequence $Q^{\tau_k \cdot a}$ to output a sequence of the form $\top \cdots \top \bot$. Here, each $\top$ or $\bot$ is an output of $\mathcal{A}_{PQ}$ (Algorithm 1). The probability that $\mathcal{A}_2$ outputs the sequence $r$ can be upper-bounded by the probability of getting $\top$ on the $k^{th}$ call to $\mathcal{A}_{PQ}$. Formally,

$$\Pr(\mathcal{A}_2(\Pi, Q^{\tau_k \cdot a}) = r) \le \Pr(\mathcal{A}_{PQ}(\Pi, \tau_k, \hat{\theta}, \varepsilon') = \top) \tag{26}$$

$\forall a$, Using $count_\tau(\Pi) < c_{min} < \theta$, (since $\theta = c_{min}/p_{min}$), Lemma 5.2, and equation 26,

$$\Pr(\mathcal{A}_2(\Pi, Q^{\tau_k \cdot a}) = r) \le \delta' \tag{27}$$

Combining equation 25 and equation 27, we get,

$$\Pr(\mathcal{A}(\Pi) = \tau_k) < \delta' \sum_{a \in A} \Pr(E_{\tau_k \cdot a} | \Pi) = \delta' \Pr(E_{\tau_k} | \Pi) \tag{28}$$

$\square$

**Lemma A.2.** *For any trajectory prefix $\tau_k = (s_1, a_1, ..., s_k, a_k, s_{k+1}), k < L$, and neighboring expert sets $\Pi, \Pi' \subseteq \Pi_{p_{min}}$, if $count_{\tau_k}(\Pi) \ge c_{min}$ then,*

$$\Pr(\mathcal{A}(\Pi) = \tau_k) \le e^{2\varepsilon'} \Pr(\mathcal{A}(\Pi') = \tau_k) + \Pr(E_{\tau_k} | \Pi)\delta'$$

*where $\varepsilon', \delta'$ are as defined in Algorithm 2.*

*Proof.* Let $r = \top \cdots \top \bot \in \{\top, \bot\}^{k+1}$.

$$\Pr(\mathcal{A}(\Pi) = \tau) = \sum_{a \in A} \left[ \Pr(\mathcal{A}_2(\Pi, Q^{\tau \cdot a}) = r^{k+1}) \Pr(E_{\tau \cdot a} | \Pi) \right] \tag{29}$$

Let $A' = \{a : a \in A, count_{\tau_k \cdot a}(\Pi) \ge c_{min}\}, A'' = A \setminus A'$.
$\forall a \in A''$,

$$count_{\tau_k \cdot a}(\Pi) = \sum_{i=1}^m [(\Pi_{j=1}^k \pi_i(a_j | s_j)) \times \pi_i(a | s_{k+1})] \ge p_{min} \times \sum_{i=1}^m [\Pi_{j=1}^k \pi_i(a_j | s_j)] = p_{min} \times count_{\tau_k}(\Pi) \tag{30}$$

$$p_{min} \times count_{\tau_k}(\Pi) \le count_{\tau_k \cdot a}(\Pi) < c_{min} \tag{31}$$

$$\Rightarrow count_{\tau_k}(\Pi) < \frac{c_{min}}{p_{min}} = \theta \tag{32}$$

The output of $\mathcal{A}_2$ is the output of a sequence of calls to $\mathcal{A}_{PQ}$ (Algorithm 1). The probability that $\mathcal{A}_2$ outputs the sequence $r$ can be upper-bounded by the probability of getting $\top$ on the $k^{th}$ call to $\mathcal{A}_{PQ}$. Using this fact along with equation 32 and Lemma 5.2, we can say,

$$\Pr(\mathcal{A}_2(\Pi, Q^{\tau_k \cdot a}) = r) \le \Pr(\mathcal{A}_{PQ}(\Pi, \tau_k, \hat{\theta}, \varepsilon') = \top) < \delta' \tag{33}$$

$$\Pr(\mathcal{A}(\Pi) = \tau_k) = \sum_{a \in A} [\Pr(\mathcal{A}_2(\Pi, Q^{\tau_k \cdot a}) = r) \Pr(E_{\tau_k \cdot a}|\Pi)] \tag{34}$$

$$= \sum_{a \in A'} [\Pr(\mathcal{A}_2(\Pi, Q^{\tau_k \cdot a}) = r) \Pr(E_{\tau_k \cdot a}|\Pi)]$$

$$+ \sum_{a \in A''} [\Pr(\mathcal{A}_2(\Pi, Q^{\tau_k \cdot a}) = r) \Pr(E_{\tau_k \cdot a}|\Pi)] \tag{35}$$

$$= \sum_{a \in A'} [\Pr(\mathcal{A}_2(\Pi, Q^{\tau_k \cdot a}) = r) \Pr(E_{\tau_k \cdot a}|\Pi)]$$

$$+ \delta' \sum_{a \in A''} \Pr(E_{\tau_k \cdot a}|\Pi) \qquad \text{[Using equation 33]} \tag{36}$$

$$\leq e^{2\varepsilon'} \sum_{a \in A'} [\Pr(\mathcal{A}_2(\Pi', Q^{\tau_k \cdot a}) = r) \Pr(E_{\tau_k \cdot a}|\Pi')]$$

$$+ \delta' \Pr(E_{\tau_k}|\Pi) \qquad \text{[Using: } \mathcal{A}_2 \text{ is } (\varepsilon', 0)\text{-DP and Lemma 5.1]} \tag{37}$$

$$\leq e^{2\varepsilon'} \sum_{a \in A} [\Pr(\mathcal{A}_2(\Pi', Q^{\tau_k \cdot a}) = r) \Pr(E_{\tau_k \cdot a}|\Pi')] + \delta' \Pr(E_{\tau_k}|\Pi) \tag{38}$$

$$\Pr(\mathcal{A}(\Pi) = \tau_k) \leq e^{2\varepsilon'} \Pr(\mathcal{A}(\Pi') = \tau_k) + \Pr(E_{\tau_k}|\Pi)\delta' \tag{39}$$

$\square$

**Lemma A.3.** *For any trajectory prefix $\tau_L$, where $L$ is the maximum length of a trajectory, and a pair of neighboring expert sets $\Pi, \Pi' \subseteq \Pi_{p_{min}}$, if $count_{\tau_L}(\Pi) < c_{min}$ then,*

$$\Pr(\mathcal{A}(\Pi) = \tau_L) < \Pr(E_{\tau_L}|\Pi) \cdot \delta'. \tag{40}$$

*If $count_{\tau_L}(\Pi) \geq c_{min}$ then,*

$$\Pr(\mathcal{A}(\Pi) = \tau_L) \leq e^{2\varepsilon'} \Pr(\mathcal{A}(\Pi') = \tau_L) \tag{41}$$

*Proof.* Let $\gamma^{(L)} = \top^L$.

$$\Pr(\mathcal{A}(\Pi) = \tau_L) = \Pr(\mathcal{A}_2(\Pi, Q_L^\tau) = \gamma^{(L)}) \Pr(E_{\tau_L}|\Pi) \tag{42}$$

If $count_{\tau_L}(\Pi) \geq c_{min}$, then using the fact that $\mathcal{A}_2$ is $(\varepsilon', 0)$-DP and Lemma 5.1, we can say that,

$$\Pr(\mathcal{A}(\Pi) = \tau_L) \leq e^{2\varepsilon'} \Pr(\mathcal{A}(\Pi') = \tau_L) \tag{43}$$

Recall that the output of $\mathcal{A}_2$ is the output of a sequence of calls to $\mathcal{A}_{PQ}$ (Algorithm 1). The probability that $\mathcal{A}_2$ outputs the sequence $r$ can be upper-bounded by the probability of getting $\top$ on the $L^{th}$ call to $\mathcal{A}_{PQ}$. Formally,

$$\Pr(\mathcal{A}_2(\Pi, Q^{\tau_L}) = \gamma^{(L)}) \leq \Pr(\mathcal{A}_{PQ}(\Pi, \tau_L, \hat{\theta}, \varepsilon') = \top) \tag{44}$$

If $count_{\tau_L}(\Pi) < c_{min}$, then using $count_\tau(\Pi) < c_{min} < c_{min}/p_{min} = \theta$, equation 44 and Lemma 5.2, we get,

$$\Pr(\mathcal{A}_2(\Pi, Q^{\tau_L}) = \gamma^{(L)}) \leq \Pr(\mathcal{A}_{PQ}(\Pi, \tau_L, \hat{\theta}, \varepsilon') = \top) < \delta' \tag{45}$$

$$\Pr(\mathcal{A}(\Pi) = \tau_L) \leq \Pr(E_{\tau_L}|\Pi)\delta' \qquad \text{[From } equation \text{ 42]} \tag{46}$$

Combining 43 and 46, we get the statement of the lemma. $\square$

### A.4 Proof of Theorem 5.3

**Theorem 5.3.** *Under Assumption 3.4, $\mathcal{A}$ is $(2\varepsilon', \frac{\delta_1}{2T})$-DP, where $\varepsilon', \delta_1$ are defined in Algorithm 2.*

*Proof.* Let $\mathcal{T}$ denote the set of all possible trajectory prefixes using $S, A$. From Lemma 5.4, we have, $\forall \tau \in \mathcal{T}$, and all pairs of neighboring sets of experts $\Pi, \Pi' \subseteq \Pi_{p_{min}}$,

$$\Pr(\mathcal{A}(\Pi) = \tau) \leq e^{2\varepsilon'} \Pr(\mathcal{A}(\Pi') = \tau) + \Pr(E_\tau | \Pi) \cdot \delta' \tag{47}$$

where $\delta' = \delta_1/(2LT)$.
To prove that $\mathcal{A}$ is $(2\varepsilon', \delta_1/(2T))$-differentially private, we need to prove the following for any pair of neighbouring $\Pi, \Pi' \subseteq \Pi_{p_{min}}$:

$$\sum_{\tau \in \mathcal{T}} \max\{\Pr(\mathcal{A}(\Pi) = \tau) - e^{2\varepsilon'} \Pr(\mathcal{A}(\Pi') = \tau), 0\} \leq \frac{\delta_1}{2T} \tag{48}$$

Note that, for any fixed $k$,

$$\sum_{\tau \in \{\tau : \tau \in \mathcal{T}, |\tau| = k\}} \Pr(E_\tau | \Pi) = 1 \tag{49}$$

where $|\tau|$ denotes the length of the trajectory prefix $\tau$.

$$\sum_{\tau \in \mathcal{T}} \Pr(E_\tau | \Pi) = \sum_{k=1}^{L} \left[ \sum_{\tau \in \{\tau : \tau \in \mathcal{T}, |\tau| = k\}} \Pr(E_\tau | \Pi) \right] = L \tag{50}$$

where $L$ is the length of a full trajectory. From equation 47, we have,

$$\sum_{\tau \in \mathcal{T}} \max\{\Pr(\mathcal{A}(\Pi) = \tau) - e^{2\varepsilon'} \Pr(\mathcal{A}(\Pi') = \tau), 0\} \leq \sum_{\tau \in \mathcal{T}} \Pr(E_\tau | \Pi)\delta' = L\delta' = \frac{\delta_1}{2T} \tag{51}$$

This proves Eqn 48, and hence the statement of the theorem.

$\square$

## B EXPERIMENTAL DETAILS

### B.1 Dataset Generation

We set $p_{min} = 0.02$ for all the environments.

**LunarLander:** Among the various variable which govern the dynamics of this environment, we chose to modify gravity (default = 9.8), wind power (default = 0.0) and turbulence (default = 0.0) to train different experts. We modified gravity over 10 equally spaced values from 9.0 to 11.0, wind power over 10 equally spaced values between 0.0 to 5.0, and similarly turbulence over 10 equally spaced values from 0.0 to 0.5, to generate 1000 environment settings. For each of these environment settings, we trained agents using the PPO algorithm for $1e6$ updates along with hyperparameter tuning of learning rate and batch size. Each agent is a neural network with hidden-layers of sizes (512, 256, 64). For each environment setting, we chose the top 3 agents that performed the best on *the corresponding environment*, to form our 3000 experts.

**CartPole:** For CartPole, we chose to vary gravity, the magnitude of the force exerted when the cart is pushed and the mass of the cart. The default values of gravity, force magnitude and mass of the cart are 9.8, 10.0 and 1.0 respectively. We varied each control over 10 equally spaced values over a fixed range. For gravity, we fixed this range as (8.75, 11.0), for force magnitude as (9.0, 11.25) and for mass of the cart as (0.8, 1.25). Similar to above, for each of these 1000 environment settings, we trained agents using the PPO

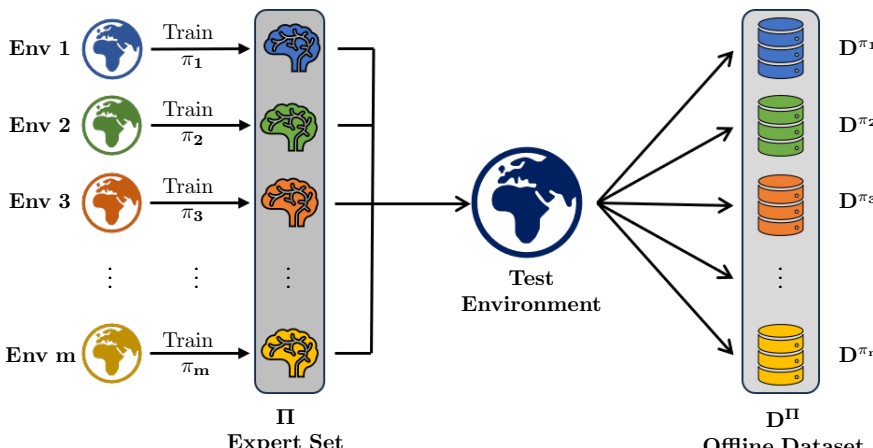

Figure 5: Outline of Data Generation scheme; We train experts on multiple environments as described below.

algorithm for $1e6$ updates while tuning the learning rate and batch size. Each agent is a neural network with hidden-layers of sizes (512, 256, 64). We chose the top 3 agents from each setting to form our 3000 experts.

**Acrobot:** For this environment, we varied the lengths of the 2 connected links and their masses. The default values of all these parameters is 1.0. We assign the same length to both the links in each setting. We vary this length, the mass of link 1 and the mass of link 2 over 10 equally spaced values across fixed ranges for each. We fix these ranges as (0.8, 1.2) for the length of the links, (0.9, 1.1) for the mass of link 1 and (0.9, 1.1) for the mass of link 2. The combination of these gives us 1000 environment settings. Again, we trained agents using PPO for $1e6$ updates over multiple learning rates and batch sizes and picked the top 3 agents from each setting to form our 3000 experts. Each agent is a neural network with hidden-layers of sizes (512, 256, 64).

**HIV Treatment:** In this domain, the 6-dimensional continuous state represents the concentrations of various cells and viruses. The actions correspond to the dosages of two drugs. The reward is a function of the health outcomes of the patient, and it also penalises the side-effects due to the quantity of drugs prescribed. Hence, the default reward function has 2 penalty terms, a penalty of 20000 on the dosage of drug 1, and a penalty of 2000 on the dosage of drug 2 (Ernst et al., 2006). In the default setting of the environment, we assume that the treatment at each steps is applied for 20 days and that there are 50 time steps in an episode. To train experts, we vary the penalty on drug 1 over 50 equally spaced values from 18750 to 21250, and the penalty on drug 2 over 20 equally spaced values from 1850 to 2150. Another variation that we introduce is whether the efficacy of the administered drug is noisy or not. So we either add no noise to the drug efficacy, or we add gaussian noise with standard deviation 0.01. All these modifications give us 2000 different environment settings. We trained agents using PPO for $2e6$ updates on each of the environment settings and over multiple learning rates and batch sizes. Each agent is represented by a neural network with a single hidden layer of size 512. We then picked the top agents from each of the 2000 environment settings to create experts. We picked 1000 more top-performing agents from the 2nd best agents of all the environment settings to get 3000 experts in total.

Once we have chosen 3000 experts for any of the above environments, we modify each expert's policy so that for each state, the action with the highest probability gets $(1 - (|A| - 1) \cdot p_{min})$ probability (where $|A|$ denotes the total number of actions), whereas all the other actions get $p_{min}$ probability of being executed. We then use these modified experts to sample 20 trajectories each from the default environment to form our offline dataset.

## B.2 Expert-Level DP-SGD

We adapted DP-SGD to the expert-level privacy setting to form the baseline that we compare against. To achieve this, during gradient updates, we ensure that an expert contributes at most one transition to a batch. To construct a batch of size $b$, we randomly sample $b$ experts from the total $m$ experts, and then pick a transition from each selected expert. This is analogous to DP-SGD training on a dataset of size $m$. For privacy accounting, we use PLD accountant from the `tensorflow_privacy` library.

---

**Algorithm 4** Expert-Level DP-SGD (outline)

---

**Require:** Model Parameters $\phi$, Expert Datasets $D^\Pi$, Expert Policies $\Pi$, Loss function $\mathcal{L}(\phi) = \frac{1}{b}\sum_{i=1}^{b}\mathcal{L}(\phi, x_i)$, Learning Rate $\eta$, Batch size $b$, Noise parameter $\sigma$, Clipping threshold $C$, Training steps $N$

1: Initialize $\phi_0$ randomly
2: **for** $t = 1, \ldots, N-1$ **do**
3:     Sample subset of experts $B$ from $\Pi$ with subsampling probability $\frac{b}{m}$
4:     For each $\pi_i \in B$ sample transition $x = (s, a, r, s')$ from $D^{\pi_i}$ to form a batch of transitions $B_{trans}$
5:     For each $x_i \in B_{trans}$ compute gradients $g_i(\phi_t) \leftarrow \nabla_{\phi_t}\mathcal{L}(\phi_t, x_i)$
6:     Scale gradients to fit clipping threshold $\bar{g}_i(\phi_t) \leftarrow g_i(\phi_t)/\max\left(1, \frac{\|g_i(\phi_t)\|_2}{C}\right)$
7:     Add gaussian noise to gradients $\tilde{g}_t \leftarrow \frac{1}{b}\left(\sum_{i=1}^{b}\bar{g}_i(\phi_t) + \mathcal{N}(0, \sigma^2 C^2 \mathrm{I})\right)$
8:     Update model parameters $\phi_{t+1} \leftarrow \phi_t - \eta \cdot \tilde{g}_t$
9: **return** $\phi_T$

---

Algorithm 4 describes a general outline for expert-level DP-SGD which we use as a baseline to compare our method against. The loss function here corresponds to the offline RL algorithm.

In our method, Algorithm 4 is incorporated in the training scheme in Algorithm 3 (lines 5 and 6) in the main paper. At each training step, with probability $p$, we execute lines 3-8 in Algorithm 4 to train on a batch of transitions. Otherwise, we sample a batch of transitions from $\mathcal{D}_{stable}^\Pi$ and perform a normal SGD update (this has no privacy cost). The privacy analysis for Algorithm 3 proceeds in a manner similar to that in standard DP-SGD (Abadi et al., 2016) with the exception of the subsampling probability which must be modified to be $q = p \cdot \frac{b}{m}$ (in the analysis only) to account for the randomness in picking the unstable dataset for sampling.

## B.3 Hyperparameter Tuning

For all the environments, we assume that $p_{min} = 0.02$. For LunarLander, CartPole and HIVTreatment, for $\varepsilon = 5$, we set $\varepsilon_1 = 0, \varepsilon_2 = \varepsilon, \delta_1 = 0, \delta_2 = \delta$ and resort to pure expert-level DP-SGD training since the privacy budget is not enough to get meaningful stable trajectories for training. For Acrobot, we set $\varepsilon_1 = \varepsilon, \varepsilon_2 = 0, \delta_1 = \delta, \delta_2 = 0$. For all other cases, we set $\varepsilon_1 = \frac{3\varepsilon}{4}, \varepsilon_2 = \frac{\varepsilon}{4}, \delta_1 = \frac{9\delta}{10}, \delta_2 = \frac{\delta}{10}$.

We use Conservative Q-Learning (CQL) (Kumar et al., 2020) and Discrete Batch-Constrained Deep Q-Learning (BCQ) (Fujimoto et al., 2019a) for training on the offline data. For Lunar Lander, Acrobot and CartPole, we fix all neural network hidden layer sizes to (256, 256). For HIV Treatment, we use a neural network of one hidden layer of size 512 to represent the Q-function. For DP-SGD updates, we clip all gradient norms to 1.0. For Acrobot, for $\varepsilon = 10, 7.5, 5$, we set $T$ (defined in Algorithm 2) equal to 25, 20, 20 respectively. For LunarLander, for $\varepsilon = 10, 7.5$, we set $T$ equal to 100 and 62 respectively. For CartPole, for $\varepsilon = 10, 7.5$, we set $T$ equal to 25 and 20 respectively. We use $T = 50$ for HIVTreatment.

We tune the following hyperparameters for our method: learning rate $\eta$, batch size $b$, probability of sampling from $\mathcal{D}_{unst}^\Pi$ during training ($p$), and DP-SGD noise multiplier ($s$) which is the standard deviation of the gaussian noise applied to the gradients divided by the clipping threshold. We perform a grid search over all hyper-parameters. The search spaces for different hyperparameters are as follows: $\eta$ - [0.0001, 0.0005, 0.001, 0.005, 0.01], $b$ - [64, 128, 256], $p$ - [0.9, 0.8, 0.5, 0.0], and $s$ - [10.0, 20.0, 30.0, 40.0, 50.0, 80.0]. For each

configuration of hyperparameters, we let the model train for as long as possible with the given value of $\varepsilon_2$. We report the average episodic return obtained over 10 evaluation runs spaced over the last 10000 steps of training. The best hyper-parameter configurations for each setting are given in Table 3.

Table 3: Best hyper-parameter settings across different environments, offline-RL algorithms and $\varepsilon$ values.

| Environment | $\varepsilon$ | CQL | | | | BCQ | | | |
|---|---|---|---|---|---|---|---|---|---|
| | | $\eta$ | $b$ | $p$ | $s$ | $\eta$ | $b$ | $p$ | $s$ |
| **Acrobot** | 5 | 0.001 | 128 | 0.0 | - | 0.01 | 256 | 0.0 | - |
| | 7.5 | 0.001 | 128 | 0.0 | - | 0.01 | 256 | 0.0 | - |
| | 10 | 0.0005 | 128 | 0.0 | - | 0.01 | 256 | 0.0 | - |
| **LunarLander** | 5 | 0.0005 | 256 | 1.0 | 80 | 0.005 | 64 | 1.0 | 10 |
| | 7.5 | 0.005 | 64 | 0.8 | 20 | 0.001 | 128 | 0.5 | 50 |
| | 10 | 0.0005 | 64 | 0.8 | 40 | 0.005 | 128 | 0.5 | 20 |
| **CartPole** | 5 | 0.005 | 256 | 1.0 | 20 | 0.005 | 128 | 1.0 | 10 |
| | 7.5 | 0.001 | 128 | 0.9 | 50 | 0.01 | 128 | 0.5 | 20 |
| | 10 | 0.0001 | 128 | 0.8 | 80 | 0.001 | 128 | 0.5 | 40 |
| **HIVTreatment** | 5.0 | 0.01 | 256 | 1.0 | 10 | 0.005 | 128 | 1.0 | 20 |
| | 7.5 | 0.0001 | 64 | 0.0 | - | 0.005 | 64 | 0.9 | 20 |
| | 10 | 0.0001 | 64 | 0.0 | - | 0.005 | 64 | 0.9 | 20 |

