# OpenReview forum: "Preserving Expert-Level Privacy in Offline Reinforcement Learning"
_TMLR — Accepted by TMLR_

### Review · Reviewer_f6f8 · 2025-07-16

**Summary Of Contributions:**

This paper introduces a novel and timely contribution to privacy-preserving offline reinforcement learning (RL). It addresses the underexplored challenge of expert-level privacy—protecting the sensitive strategies of individual experts contributing to an offline dataset—with a theoretically grounded and empirically validated approach. The work is well-motivated, methodologically sound, and offers practical advancements.

Problem Formulation: Formalizes expert-level differential privacy (DP) for offline RL, where the goal is to protect all trajectories contributed by a single expert, a coarser-grained notion than trajectory-level DP. This is motivated by real-world applications (e.g., healthcare, advertising) where experts (e.g., doctors, advertisers) require privacy.

Algorithm: Proposes a two-stage consensus-based framework:

Stage 1 (Data Filtering): Uses a privatized variant of the Sparse Vector Technique (SVT) to identify "stable" trajectory prefixes (common across many experts) that can be used in training without noise.

Stage 2 (Selective DP-SGD): Applies expert-level DP-SGD only to "unstable" data, reducing overall noise injection.

Theoretical Guarantees: Rigorously proves $(\epsilon,\delta)-DP for the entire pipeline, leveraging advanced composition and properties of SVT.

Empirical Validation: Demonstrates superior performance over an expert-level DP-SGD baseline across four RL benchmarks (CartPole, Acrobot, LunarLander, HIV Treatment), especially for higher $\epsilon$ (lower privacy). The approach integrates seamlessly with standard offline RL algorithms (CQL, BCQ).

**Audience:**

Yes

**Broader Impact Concerns:**

I do not have any concerns.

**Claims And Evidence:**

Yes

**Requested Changes:**

This paper compares the definition of expert level privacy and trajectory level privacy. I think that these concepts are close to "user level privacy" and "item level privacy". Therefore I suggest to discuss these in related works.

[1] Levy, Daniel, et al. "Learning with user-level privacy." Advances in Neural Information Processing Systems 34 (2021): 12466-12479.

[2] Zhao, Puning, et al. "A huber loss minimization approach to mean estimation under user-level differential privacy." Advances in Neural Information Processing Systems 37 (2024): 130018-130056.

[3] Zhou, Mingxun, et al. "Locally differentially private sparse vector aggregation." 2022 IEEE Symposium on Security and Privacy (SP). IEEE, 2022.

**Strengths And Weaknesses:**

Strengths:

Novelty: First to formalize and address expert-level privacy in offline RL, filling a gap in prior work (e.g., trajectory-level DP or tabular/linear settings).

Practical Flexibility: Compatible with any gradient-based offline RL algorithm and general function approximators (e.g., DNNs), enhancing real-world applicability.

Theoretical Soundness: Provides rigorous privacy guarantees via a non-trivial adaptation of SVT, addressing dependencies between queries and the private database (experts).

Compelling Experiments: Extensive evaluation on diverse tasks shows significant gains over DP-SGD (e.g., +20–40% normalized return at
$\epsilon=10$). Ablation on expert count (Fig. 4) validates scalability.

Clarity: Well-structured, with clear problem setup, algorithm pseudocode, and intuitive explanations.

Weaknesses:

The assumption may be relaxed. I suggest to explore extensions to continuous actions (e.g., via density estimation) and behavior-cloned experts in future work.

Is the theoretical guarantee optimal? I do not see any further analysis.

Assumption 3.6 is too restrictive. Assuming minimum action probability is not realistic, as in real scenarios, some actions are only taken limited times. However, I think for a theoretical analysis I think that it is OK for the current paper.

---

> ### Author Response · Authors · 2025-07-31
> **Response to Reviewer f6f8**
>
> We thank the reviewer for their feedback and address their concerns below.
>
> > The assumption may be relaxed. I suggest to explore extensions to continuous actions (e.g., via density estimation) and behavior-cloned experts in future work.
>
> We thank the reviewer for their suggestions. We will add these points to the future work section of the paper and look forward to working on these in future work.
>
> > Is the theoretical guarantee optimal? I do not see any further analysis.
>
> We do not make any claims about the optimality of the theoretical bounds themselves in our work. To understand and analyze such bounds, a privacy audit needs to be carried out. Auditing privacy for offline RL settings is, to the best of our knowledge, an unstudied problem, and an exciting direction for future work. However, it is beyond the scope of this work.
>
> > Assumption 3.6 is too restrictive. Assuming minimum action probability is not realistic, as in real scenarios, some actions are only taken limited times. However, I think for a theoretical analysis I think that it is OK for the current paper.
>
> We acknowledge that our algorithm is only meaningful for classes of policies for which $p_{min} > 0$, and this knowledge may not be available in several scenarios.
> Currently, $p_{min}$ is necessary to lower-bound the probability ratio that arises from our differential privacy analysis (equations 30-32 in Section A.3 of the appendix). It is non-trivial to remove this dependency, and it is out of scope for this paper. However, we look forward to addressing this dependency in future work.
>
> > This paper compares the definition of expert level privacy and trajectory level privacy. I think that these concepts are close to "user level privacy" and "item level privacy". Therefore I suggest to discuss these in related works.\
> [1] Levy, Daniel, et al. "Learning with user-level privacy." Advances in Neural Information Processing Systems 34 (2021): 12466-12479.\
> [2] Zhao, Puning, et al. "A huber loss minimization approach to mean estimation under user-level differential privacy." Advances in Neural Information Processing Systems 37 (2024): 130018-130056.\
> [3] Zhou, Mingxun, et al. "Locally differentially private sparse vector aggregation." 2022 IEEE Symposium on Security and Privacy (SP). IEEE, 2022.
>
> We thank the reviewer for pointing out relevant work and will be sure to reference them in the updated paper.
>
> As they have correctly pointed out, the relation between trajectory-level and expert-level DP discussed in our work is analogous to the relationship between item-level and user-level DP in existing privacy literature. We use alternate terminology (trajectory-level, expert-level etc.) to convey what items and users refer to in an offline RL setting.
>
> We also point the reviewer to the discussion on user-level privacy in the response to the 4th question raised by Reviewer CcoS.

---

### Review · Reviewer_CcoS · 2025-07-22

**Summary Of Contributions:**

This paper introduces and formalizes the setting of expert-level differential privacy in offline reinforcement learning (RL), where the goal is to learn a high-utility policy while protecting the identity and strategies of contributing experts. The authors propose a two-stage algorithm that first filters privacy-safe trajectory prefixes using a novel variant of the Sparse Vector Technique (SVT) and then trains any gradient-based offline RL algorithm via a selectively private SGD, where only the privacy-sensitive portions are noised. Theoretical results establish (ε, δ)-DP guarantees under expert-level neighboring datasets, and empirical evaluations on multiple RL benchmarks demonstrate improved utility compared to baseline DP-SGD methods adapted to expert-level privacy.

**Audience:**

Yes

**Broader Impact Concerns:**

None.

**Claims And Evidence:**

Yes

**Requested Changes:**

One of my major concerns is the motivation of the expert-level DP in offline RL. First, I am not fully convinced by Example on page 5. That is, why we need to protect the privacy of the doctors (if I understand it correctly). Second, it seems to me that the key difference between expert-level and trajectory-level is similar to the difference between user-level and item-level DP. That is, expert-level is similar to user-level in that changing one expert/user, will change multiple data points (trajectories). If this is correct, then my question is -- can we leverage the recent advance in user-level DP optimization (like supervised learning or stochastic optimization) to handle expert-level offline RL?

I am looking forward to seeing the comments from the authors on the above questions.

Minor issue: "To account for privacy amplification while sampling batches from the dataset, we use the use the
fraction" it has two "use the"

**Strengths And Weaknesses:**

**Strengths**
+ The paper proves (ε, δ)-DP under the expert-level neighbor definition, carefully accounting for the challenges of streaming queries dependent on the private dataset. To me, the privacy analysis appears to be rigorous and non-trivial.

+ The proposed framework can be combined with any gradient-based offline RL algorithm, increasing its practical impact and adaptability.


**Weaknesses**

+ The paper lacks theoretical bounds on the suboptimality of the learned policy under the DP constraints, which would make the privacy-utility tradeoff more explicit.

+ The current privacy analysis relies on several assumptions, some of which may affect the generalizability of the proposed method.

---

> ### Author Response · Authors · 2025-07-31
> **[1/3] Response to Reviewer CcoS**
>
> We thank the reviewer for their feedback and address their concerns below.
>
> > The paper lacks theoretical bounds on the suboptimality of the learned policy under the DP constraints, which would make the privacy-utility tradeoff more explicit.
>
> We provide fine-grained privacy guarantees by retaining as much noiseless data as possible during the filtering step, rather than uniformly adding noise to all gradients. With the resulting dataset, we will inherit the utility guarantees of the downstream offline RL algorithm, based on the coverage properties of the filtered data, as we discuss in the section ‘Sub-optimality guarantees for the learned policy’ on page 7. The analysis of the distributional properties of this dataset, however, is highly non-trivial and problem-dependent and we leave this treatment to future work.
>
> However, we perform an extensive experimental evaluation to show that our algorithm retains much of the utility of the non-private baseline algorithm, consistently outperforming DP-SGD.
>
> > The current privacy analysis relies on several assumptions, some of which may affect the generalizability of the proposed method.
>
> We comment on the different assumptions made in our work below.
>
> - **Assumption 3.4**: Discrete action space.\
> This is a fairly mild assumption satisfied in many natural settings.
>
> - **Assumption 3.5**: Query access to expert policies.\
> An example of an important setting where this assumption is reasonable to make is when a student model needs to be learned from an ensemble of teacher models, in a privacy preserving manner. In such a setting, these teacher models form our expert policies. The PATE (Private Aggregation of Teacher Ensembles) method introduced in [1] also makes a similar assumption, when it assumes query access to teacher models.\
> In future work, we want to make our method more applicable by relaxing this assumption. We believe that it might be possible to replace expert policies by their behavior-cloned versions (by training each on the given private data), with a slight loss in utility. However, proving this is out of scope for the current paper.
>
> - **Assumption 3.6**: Access to minimum action probability ($p_{min}$).\
> For our algorithm to be meaningful $p_{min} > 0$, for which some knowledge of the policy class is required. This is possible in some settings, like the task of learning from private teacher models discussed above. We acknowledge that this somewhat restricts the generalizability of our method. Currently, $p_{min}$ is necessary to lower-bound the probability ratio that arises from our differential privacy analysis (equations 30-32 in Section A.3 of the appendix). It is non-trivial to remove this dependency, and it is out of scope for this paper. We look forward to removing the dependency on $p_{min}$ in future work.
>
>
> While our method assumes the above, we would like to highlight that we remove a number of highly restrictive assumptions made by existing work in private RL (such as tabular settings, linear function approximators etc.). Our work is practically relevant as the first private offline RL method which allows the use of any sophisticated, gradient-based offline RL algorithm with general function approximators and continuous state spaces, and is thus of interest to the privacy and broader reinforcement learning communities.
>
> [1] Papernot, Nicolas, et al. "Scalable private learning with pate." arXiv preprint arXiv:1802.08908 (2018).

---

> ### Author Response · Authors · 2025-07-31
> **[2/3] Response to Reviewer CcoS**
>
> > One of my major concerns is the motivation of the expert-level DP in offline RL. First, I am not fully convinced by Example on page 5. That is, why we need to protect the privacy of the doctors (if I understand it correctly).
>
> In the example stated in page 5, the privacy of the treatment methodologies may be considered sensitive information in some scenarios. Notably, in the case of medical centres and research institutes where patients undergo experimental, and often proprietary, treatment techniques, the exact treatment paradigm may not be released publicly. This motivates the need for expert-level privacy in this setting.
>
> We also propose another example to motivate our problem. Consider the case of financial portfolio management. Different fund managers employ proprietary and privacy-sensitive investment strategies that guide their trading decisions. In such a setting, treating individual fund managers as experts, learning an expert-level private investment strategy (policy) using offline RL will be beneficial as a starting strategy for new investors and other parties which do not have specialized investment strategies.
>
> Alternatively, we may consider the case of political campaign optimization in a multi-party democracy. Political parties may use proprietary algorithms to maximize their reach and voter impact, whereas independent candidates, lacking the same amount of resources, are placed at a significant disadvantage. Access to a privately learnt aggregate campaigning strategy will allow such players to compete on a somewhat equal footing, allowing voters greater effective choice.
>
> At a high-level, this method aims to solve the ‘warm start’ problem for new players, when all the available data is private and cannot be directly trained on.
>
> We would be happy to add this discussion and the additional motivating examples to future revisions of our work.
>
>
> > Second, it seems to me that the key difference between expert-level and trajectory-level is similar to the difference between user-level and item-level DP. That is, expert-level is similar to user-level in that changing one expert/user, will change multiple data points (trajectories). If this is correct, then my question is -- can we leverage the recent advance in user-level DP optimization (like supervised learning or stochastic optimization) to handle expert-level offline RL?
>
> Most practical approaches for achieving user-level differential privacy are based on clipping user-level gradients to bound user contributions [1, 2, 3, 4]. In fact, to the best of our knowledge, these are the only approaches that have been empirically tested to be successful on large scale datasets. The baseline used for comparison in our paper (described in Appendix B.2) is a representative of this class of approaches, and a natural way to tackle our problem statement. We demonstrate that our method significantly outperforms this baseline, since it identifies a subset of the data where noisy training is not required, before running the same baseline on the rest of the data.
>
> Note that our method can also be combined with any other similar user-level differentially private algorithm (adapted to the RL setting) which can be used for training on the unstable data instead of the baseline proposed by us. The main novel contribution of our algorithm is the creation of the stable dataset which can be used for training with any non-private RL algorithm.
>
> Other approaches for achieving user-level differential privacy are mostly theoretical in nature and come with restrictions that limit their practical usage, for example, restrictions on the function class. On the other hand, our method can be used with any general function approximators and continuous, high-dimensional state spaces. The main motivation behind this work is to create an expert-level differentially private algorithm for solving offline deep-RL tasks.
>
> We will be happy to update the paper with this discussion.
>
> [1] Daniel Levy, Ziteng Sun, Kareem Amin, Satyen Kale, Alex Kulesza, Mehryar Mohri, and
> Ananda Theertha Suresh. Learning with User-Level privacy. Advances in Neural Information
> Processing Systems, 34:12466–12479, 2021.
>
> [2] Raef Bassily and Ziteng Sun. User-level Private Stochastic Convex Optimization with Optimal
> Rates. In International Conference on Machine Learning, pages 1838–1851. PMLR, 2023.
>
> [3] Hilal Asi and Daogao Liu. User-level Differentially Private Stochastic Convex Optimization:
> Efficient Algorithms with Optimal Rates. In International Conference on Artificial Intelligence
> and Statistics, pages 4240–4248. PMLR, 2024.
>
> [4] Soham De, Leonard Berrada, Jamie Hayes, Samuel L Smith, and Borja Balle. Unlocking high-accuracy differentially private image classification through scale. arXiv preprint
> arXiv:2204.13650, 2022.

---

> ### Author Response · Authors · 2025-07-31
> **[3/3] Response to Reviewer CcoS**
>
> > Minor issue: "To account for privacy amplification while sampling batches from the dataset, we use the use the fraction" it has two "use the"
>
> We thank the reviewer for pointing out this typographical error, and will be sure to correct it in future drafts.

---

### Review · Reviewer_rtSW · 2025-08-06

**Summary Of Contributions:**

This paper investigates scenarios for reinforcement learning (RL) with differential privacy (DP) where the thing to be kept private are the contributions from different "expert" policies which are used in the (offline) training of a new policy.

The sensitive data used by these offline RL methods take the form of *trajectories*. The contention of the paper is that if a particular trajectory (or prefix thereof) is "common enough" among the experts then using a variation of the sparse vector technique is enough to guarantee privacy. However, if the trajectory is less common then additional privacy protections are needed in the form of DP-SGD.

The main contributions are:

* formulating this kind of problem under privacy constraints
* providing a new "sorting" approach to pull out common trajectories
* some experimental evidence for the approach

I question whether this approach is truly "practical" in the way claimed by the authors, however. It seems quite restricted to schenarios where the experts are simulateable policies and not actual measured data.

**Audience:**

Yes

**Broader Impact Concerns:**

From Assumptions 3.5 and 3.6 it seems that the primary motivation is that the experts are themselves machines, making the motivating application in Example 3 incredibly problematic, especially from an ethics perspective: are we going to collect data from sepsis patients based on automated AI experts providing treatments by taking actions and observing trajectories? I don't think this is what the authors mean to suggest but providing this as a motivating application but then allowing unfettered querying of policies are not compatible. Please remove this example or provide a better example that is more compatible with the solution setup.

**Claims And Evidence:**

Yes

**Requested Changes:**

**Motivation/applications:**

* The experiments are done on 3 mechanical/robotic applications and one healthcare application. I am not sure the first three are really reflective of privacy-relevant applications, although perhaps some case could be made. This is OK: we often work with the datasets that we have. For the HIV Treatment application, since it is a simulator, is seems that the approach is not feasible with actual observational data. Perhaps the authors can clarify a bit more how one might work with actual recordings of trajectories.

* I am not sure if the assumption that rewards being deterministic is not without loss of generality. In the healthcare application, the actions and states of patients can be sufficiently different that the rewards are almost surely not deterministic: they would be patient-specific and also location specific. It's not the case that the same experts are treating the same patient in some controlled way. Some discussion on how assumptions do and do not hold in the motivation application(s) would be helpful.

* The privacy analysis assumes that the trajectories are all of the same length. This is actually fine because the trajectories experimentally are all synthetic. The authors should be clearer about whether or not this approach is intended to work with real data trajectories/traces or only with full access to simulators which can generate as many trajectories as we want.

**Mathematical setup:**

* I think formally describing a trajectory would be helpful. It's sort of implicit in Definition 3.1. In particular, it seems that $\tau \in \{ S \times A \times R \times S \}^{\ast}$ so that trajectories have infinite horizons (which makes sense for discounted MDP)? But later in Algorithm 1 $\mathcal{D} = \{ S \times A \}^\ast$ because the rewards are ignored (which is not really without loss of generality, as noted above. But then on page 7 we get $\{ S \times A \times R \times S \}^{\ast}$ again. So what is it?

* Each expert $\pi$ induces a distribution on $\mathcal{D}$, a collection $\Pi$ of experts a collection of distributions on $\mathcal{D}$. This makes the trajectory-level neighborhood and expert-level neighborhood definitions have quite different flavors: trajectory-level neighborhood seems quite strong: you would need some sort of "smoking gun" trajectory that would only be in one $D$ and not the other $D'$. Is this a practical difference to consider? It seems not from the discussion before 3.3 but it would be good for the authors to clarify.

* In practice, it seems getting the *full set of trajectories* generated by an expert $D^{\pi_i}$ is unlikely: wouldn't you only get a sample? So is $D^{\Pi}$ really the data set under consideration?

* Is the $count$ function correctly named? Should this function also include the state transition probabilities and not just the policy actions?

**Algorithms/Experiments:**

* While a lot of work goes into splitting into stable and unstable trajectory sets, it would be nice to see some sort of (experimental) verification that using the unstable set has measurable benefits, since in some cases the training is done only with the stable set.

* How is the clipping level chosen in practice?

* Is the sampling in the experiments done without replacement? There are often gaps in the implementation between privacy amplification by subsampling (in the pseudocode) and actual implementation (shuffle and process in batches).

* (Figure 3) Shouldn't $\delta << \frac{1}{m}$ instead of equal to $\frac{1}{m}$? It was not clear from Appendix B.

* I would have expected a bit more exploration of different $\epsilon$ levels than just the 3 selected. Is this because training is very expensive? If so, does that limit the practicality of this approach?

* For completeness, some evidence for how things "break down" with smaller $\epsilon$ would be useful to see (even in an appendix). It's not clear if these large $\epsilon$ values are due to the complexity of the applications or lack of data or some other factors.

**Strengths And Weaknesses:**

**Strengths:**

* As the authors point out, experimental evaluation of DP+RL methods are relatively limited.
* The modification to the SVT may be of independent interest.
* The essential structure of the problem is interesting, in that it is not quite group privacy.

**Weaknesses:**

* It is not entirely clear that the experimental settings reflect real scenarios in which privacy is an issue. The example on Page 5 of personalized healthcare is interesting but the closest experiment to this setting is based on synthetic agents, making the privacy question less clear.
* Some of the mathematical setup is a little unclear, especially since some assumptions (like deterministic awards) are not without loss of generality.

---

> ### Author Response · Authors · 2025-08-12
> **[1/5] Response to Reviewer rtSW**
>
> We thank the reviewer for their thorough review of our work and for the several relevant and insightful questions/concerns they have posed. We address them in detail below:
>
> > It is not entirely clear that the experimental settings reflect real scenarios in which privacy is an issue. The example on Page 5 of personalized healthcare is interesting but the closest experiment to this setting is based on synthetic agents, making the privacy question less clear. \
> The experiments are done on 3 mechanical/robotic applications and one healthcare application. I am not sure the first three are really reflective of privacy-relevant applications, although perhaps some case could be made. This is OK: we often work with the datasets that we have. For the HIV Treatment application, since it is a simulator, is seems that the approach is not feasible with actual observational data. Perhaps the authors can clarify a bit more how one might work with actual recordings of trajectories.
>
> We first note that we do not assume access to the environment simulator when learning the private policy from the offline dataset. Our method is indeed designed to work with real trajectory recordings, per the offline RL setting.
>
> We have used synthetic agents to generate the offline datasets that we are working with. We first trained these synthetic agents (experts) to varying degrees of optimality, as shown in Figure 2 (p. 10, Sec 6.1). The goal of using a set of experts with varying degrees of optimality is to mimic real-world settings where the underlying expert policies generating the offline data are expected to differ in behaviour.
>
> These experts are allowed to then interact with the environment (simulator), and their trajectories are logged and collected into the aggregated dataset $D^\Pi$. We can also directly start our algorithm with an already present dataset $D^\Pi$ (along with query-able policies), if available. In fact, this is the intended use of our approach.
>
> Further, the environments we use in our work are a subset of the Gymnasium environments, which have been used extensively in previous works in reinforcement learning to test the performance of learning algorithms. These have also been used to create benchmarks in the offline setting, and more recently in private offline RL works such as [1], which focus on the trajectory level notion of differential privacy.
>
> In addition to standard RL environments, we also use the HIV treatment environment which is closer to a real-world privacy-related application, as the reviewer pointed out. We note once again that our private offline RL algorithm does not require access to the simulator (environment) to run. This, in fact, breaks the “offline” setting [2, 3].  We only use offline trajectories recorded by several behavioural policies (experts), and this process will be identical when running our algorithm on real-world recordings of trajectories. We, however, do assume query-access to expert policies currently, which we aim to relax in future work.
>
> > I am not sure if the assumption that rewards being deterministic is not without loss of generality. In the healthcare application, the actions and states of patients can be sufficiently different that the rewards are almost surely not deterministic: they would be patient-specific and also location specific. It's not the case that the same experts are treating the same patient in some controlled way. Some discussion on how assumptions do and do not hold in the motivation application(s) would be helpful.
>
> We do not make the assumption that rewards are deterministic. In fact, our framework does not use the rewards in any way during the sorting step when trajectories are split into stable and unstable segments. Once the stable and unstable datasets are created, the rewards are used for gradient updates (both noised and unnoised) using any Offline RL algorithm in Algorithm 3.
>
> Since we aim to protect the expert policies themselves (ie, actions taken), the rewards (given a state-action pair) are not privacy-sensitive. The rewards are just present as auxiliary information per transition during the creation of the stable and unstable datasets. Hence, it does not matter whether the reward function is deterministic or probabilistic. The reward values available in the data are only used for Offline RL training later.
>
>
> **References:**
>
> [1] Rio, Alexandre, et al. "Differentially Private Deep Model-Based Reinforcement Learning." arXiv (2024).
>
> [2] Levine, Sergey, et al. "Offline reinforcement learning: Tutorial, review, and perspectives on open problems." arXiv (2020).
>
> [3] Prudencio, Rafael Figueiredo, et al.  "A survey on offline reinforcement learning: Taxonomy, review, and open problems." IEEE Transactions on Neural Networks and Learning Systems 35.8 (2023): 10237-10257.

---

> > ### Author Response · Authors · 2025-08-12
> > **[2/5] Response to Reviewer rtSW**
> >
> > > The privacy analysis assumes that the trajectories are all of the same length. This is actually fine because the trajectories experimentally are all synthetic. The authors should be clearer about whether or not this approach is intended to work with real data trajectories/traces or only with full access to simulators which can generate as many trajectories as we want.
> >
> > We refer the reviewer to the discussion on Page 8 (Section 5; Paragraph 1). While our privacy analysis assumes that the trajectories are of equal length denoted by L, in a setting where the trajectories have different lengths, we simply set L to be the length of the longest trajectory.
> >
> > For every trajectory, the overall privacy cost in the $\delta$ parameter scales linearly with the length of the trajectory ie, $\delta_1 \propto L\delta’$ (Line 1 in Algorithm 2; p. 6, Section 4), where $\delta’$ is the per-timestep cost and $\delta_1$ is the full cost of the trajectory. By setting $L$ to be the maximum length across all trajectories we carry out a worst-case analysis of the privacy leakage.
> >
> > We further clarify that we do not constrain trajectory lengths in our experiments. To calculate per timestep privacy budgets for the $\delta$ parameter, we simply set $L$ to be the length of the longest trajectory. We would be happy to improve this discussion in future revisions of our work.
> >
> > Our approach does not use access to environment simulators (only access to expert policies) at any step and is, in fact, designed to work with an offline dataset of trajectories, which is the setting of Offline RL [2,3]. Note that just having access to expert policies is not sufficient for generating as many trajectories as we want, since environment access is also required for that. Hence, we only have the static, offline dataset available for training.
> >
> > > I think formally describing a trajectory would be helpful. It's sort of implicit in Definition 3.1. In particular, it seems that  so that trajectories have infinite horizons (which makes sense for discounted MDP)? But later in Algorithm 1  because the rewards are ignored (which is not really without loss of generality, as noted above. But then on page 7 we get  again. So what is it?
> >
> > We would be happy to add an explicit definition of a trajectory earlier in the paper to improve readability and add the clarification that we deal with finite-horizon trajectories.
> >
> > A trajectory in offline RL is a series of transitions of the form $(s, a, r, s’)$ tuples, where the “next state” of every transition corresponds to the “state” of the next. In Section 3.3, we present a simplified version of this definition, which defines a trajectory as a sequence of states alternating with the actions taken while transitioning from one state to the next. This is to simplify notation since the rewards associated with transitions are not used in the privacy analysis or in the filtering stage of our algorithm. In the actual implementation, however, the rewards are always present with each transition and are only used while training using the offline RL algorithm in Algorithm 3.
> >
> > We would be happy to make this clearer in the updated version of the paper.
> >
> > > Each expert $\pi$ induces a distribution on $\mathcal{D}$, a collection $\Pi$ of experts a collection of distributions on $\mathcal{D}$. This makes the trajectory-level neighborhood and expert-level neighborhood definitions have quite different flavors: trajectory-level neighborhood seems quite strong: you would need some sort of "smoking gun" trajectory that would only be in one $\mathcal{D}$ and not the other $\mathcal{D’}$. Is this a practical difference to consider? It seems not from the discussion before 3.3 but it would be good for the authors to clarify.
> >
> > A crucial part of differential privacy is identifying the unit of privacy to be protected. Informally speaking, DP attempts to mathematically bound the difference in output distributions when one unit of input data is changed. This one unit of input data is the privacy unit, which is protected under DP. For trajectory-level privacy, the privacy unit is a single trajectory. For expert-level privacy, the privacy unit is every trajectory belonging to a given expert.
> >
> > We note that as the privacy unit becomes larger, it becomes harder to privatize the outputs. Informally speaking, bounding the difference in output distributions is harder when a larger fraction of the overall input dataset is changed. Thus, while trajectory-level neighbours are closer to each other than expert-level neighbours, the notion of expert-level DP is much stronger than the notion of trajectory-level DP.
> >
> > This is identical to the analogous relationship between example-level and user-level DP [4].
> >
> > **References:**
> >
> > [4] Levy, Daniel, et al. "Learning with user-level privacy." NeurIPS (2021)

---

> > > ### Author Response · Authors · 2025-08-12
> > > **[3/5] Response to Reviewer rtSW**
> > >
> > > > In practice, it seems getting the full set of trajectories generated by an expert $D^{\pi_i}$  is unlikely: wouldn't you only get a sample? So is $D^\Pi$ really the data set under consideration?
> > >
> > > The privacy-sensitive entities under consideration here are the expert policies. We don’t need the full set of trajectories generated by an expert in our approach. $D^{\pi_i}$ refers to an offline dataset (not necessarily the full set) of trajectories sampled using the policy $\pi_i$.
> > >
> > > These trajectories as used to form SVT-like queries (as discussed in Section 5) to filter out trajectory segments that don’t give away much information about the underlying policy actually generating that trajectory. The more number of trajectories we have, more SVT-like queries can be processed, but there is no requirement to do so for the full-set of all possible trajectories.
> > >
> > > > Is the count function correctly named? Should this function also include the state transition probabilities and not just the policy actions?
> > >
> > > Given a trajectory $\tau$, the count function gives the sum of probabilities of each expert traversing $\tau$. We want to use this quantity to check if enough experts are likely to generate this trajectory.
> > >
> > > The full probability of $\tau$ being generated by the given set of experts, is actually proportional to $count$ multiplied by the state-transition probabilities. We chose the current definition of $count$ because it is the factor of the full probability that can change across neighboring experts for a given trajectory. It also has the appropriate structure that we needed to prove our privacy results.
> > >
> > > We apologize for the lack of clarity on this at the beginning of Section 4 and will be happy to update the discussion in future revision.
> > >
> > > > While a lot of work goes into splitting into stable and unstable trajectory sets, it would be nice to see some sort of (experimental) verification that using the unstable set has measurable benefits, since in some cases the training is done only with the stable set.
> > >
> > > Different values of the sampling probability ($p$) gave the best numbers on different datasets, with $p = 0$ being the best for a few settings (for Acrobot and HIVTreatment) as the reviewer pointed out. To demonstrate the benefits of using the unstable set, we report performance numbers for different values of p on the LunarLander and CartPole environments below:
> > >
> > > | Environment |  \| BCQ   |   |    |  \| CQL  |      |   |      |
> > > |--|--|--|--|--|--|--|--|
> > > |     | **\|** **p = 0.0**  | **p = 0.5**  | **p = 1.0** | **\|** **p = 0.0** | **p = 0.8**  | **p = 1.0** |      |
> > > | LunarLander | **\|** 0.0 \(collapsed performance) | 0.57  | 0.54  | **\|** 0.0 \(collapsed performance) | 0.75    | 0.67    |
> > > | CartPole    |  **\|** 0.31   | 0.99    | 0.67    | **\|** 0.63  | 1.03    | 0.92 |
> > >
> > > The columns with $p = 0.5$ and $p = 0.8$ for BCQ and CQL respectively correspond to the best performance reported in Fig. 3. For these environments, the return values are quite bad on just using the stable data ($p = 0.0$), and it is combination of stable and unstable dataset that significantly outperforms the DP-SGD baseline. This indicates the usefulness of including the unstable dataset as well during the training process.
> > >
> > > > How is the clipping level chosen in practice?
> > >
> > > We fixed the gradient clipping level to 1.0 for all our experiments.

---

> > > > ### Author Response · Authors · 2025-08-12
> > > > **[4/5] Response to Reviewer rtSW**
> > > >
> > > > > Is the sampling in the experiments done without replacement? There are often gaps in the implementation between privacy amplification by subsampling (in the pseudocode) and actual implementation (shuffle and process in batches).
> > > >
> > > > $D_{stable}^{\Pi}$ is independently shuffled and batched. For $D_{unst}^{\Pi}$, batches are created by shuffling datasets of individual experts and randomly interleaving to ensure that each batch has at most one data point from each expert.
> > > >
> > > > At each step of Algorithm 3, we train on a batch from $D_{unst}^{\Pi}$ with probability p, and on a batch from $D_{stable}^{\Pi}$ otherwise. Since training on $D_{stable}^{\Pi}$ does not incur additional privacy cost, there are no gaps in the theoretical analysis and the actual implementation when $D_{stable}^{\Pi}$ is used for training.
> > > >
> > > > The reviewer is correct in pointing out the fact that gaps exist between the analysis and implementation when $D^{\Pi}_{unst}$ is used for training. In our implementation, we chose to shuffle-and-batch entries to optimize training time; randomly sampling individual batches incurs additional time cost.
> > > >
> > > > We also note that the gap between analysis and implementation in our work corresponds to the gap between privacy amplification by subsampling and shuffling. A promising line of work [5,6] aims to derive comparable theoretical amplification guarantees via shuffling. Empirically, however, the gap in utility is expected to be minimal when either shuffling or subsampling is used to amplify privacy [7].
> > > >
> > > > We thank the reviewer for prompting this discussion and would be happy to add this to the main paper.
> > > >
> > > > > (Figure 3) Shouldn't $\delta<<1/m$ instead of equal to $1/m$? It was not clear from Appendix B
> > > >
> > > > The privacy parameter $\delta$ in the definition of $(\varepsilon, \delta)$-differential privacy corresponds to a failure probability in protecting the privacy of each individual data point. In other words, $m\delta$ is the expected number of privacy leakages for a given when $m$ data points are present in the dataset. Theoretically, it suffices to select any value of $\delta < 1/m$. However, this too, can only provide probabilistic guarantees regarding a worst-case catastrophic failure in leaking the privacy of an expert. Thus, we take the liberty of selecting a limiting value of $\delta\rightarrow1/m$ for our experimental evaluation.
> > > >
> > > > **References:**
> > > >
> > > > [5] Erlingsson, Úlfar, et al. "Amplification by shuffling: From local to central differential privacy via anonymity." ACM-SIAM 2019.
> > > >
> > > > [6] Feldman, Vitaly, Audra McMillan, and Kunal Talwar. "Hiding among the clones: A simple and nearly optimal analysis of privacy amplification by shuffling." FOCS 2021.
> > > >
> > > > [7] Ponomareva, Natalia, et al. "How to dp-fy ml: A practical guide to machine learning with differential privacy." JAIR (2023)

---

> ### Author Response · Authors · 2025-08-12
> **[5/5] Response to Reviewer rtSW**
>
> > I would have expected a bit more exploration of different eps  levels than just the 3 selected. Is this because training is very expensive? If so, does that limit the practicality of this approach?
>
> We also experimented with values of $\varepsilon < 5$, specifically $\varepsilon = 2.5$. We found out that for such low $\varepsilon$ Algorithm 2 yields very little stable data, which is insufficient to meaningfully influence training. In these cases, the model primarily learns from the unstable data. Hence, setting $\varepsilon_1 = 0$ and $\varepsilon_2 = \varepsilon$ is the best that we can do for such low $\varepsilon$ values, by minimizing the noise injected by DP-SGD.
>
> This shows that our algorithm is more suited for larger $\varepsilon$ values in the range $\varepsilon \geq 5$. Nevertheless, this does not restrict its applicability, as most real-world scenarios typically permit $\varepsilon$ values up to 10.0.
>
> We would be happy to include this discussion in the revised version of the paper.
>
> > For completeness, some evidence for how things "break down" with smaller \varepsilon  would be useful to see (even in an appendix). It's not clear if these large  values are due to the complexity of the applications or lack of data or some other factors.
>
> Following from the response to the previous question, we use the Acrobot and CartPole environments to show the effects of using small \varepsilon values ($\varepsilon = 2.5, 5.0$).
>
> We observed that for $\varepsilon\leq 5.0$ for CartPole and $\varepsilon\leq2.5$ for Acrobot, it is better to allocate all the privacy budget to the DP-SGD training stage, by setting $\varepsilon_1 = 0$ and $\varepsilon_2 = \varepsilon$. This is in contrast to the observations for higher \varepsilon values where we have used $\varepsilon_1 = 0.75 * \varepsilon$ and $\varepsilon_2 = 0.25 * \varepsilon$ or $\varepsilon_1 = \varepsilon$ and $\varepsilon_2 = 0$, to benefit from the stable dataset as well.
>
> The following return values show how setting $\varepsilon_1$ to a high non-zero value for low $\varepsilon$ values does not give any benefits, and it is better to allocate the entire privacy budget for DP-SGD training:
>
> | Environment    | $\varepsilon$ | Performance with $\varepsilon_1 = 0.75 \* \varepsilon$, $\varepsilon_2 = 0.25 \* \varepsilon$ | DP-SGD |
> | -| -| -| -|
> | Acrobot (BCQ)  | 2.5   | 0.74   | 0.95   |
> | Cartpole (BCQ) | 2.5  | 0.28  | 0.58   |
> | Cartpole (BCQ) | 5.0    | 0.56  | 0.60  |
>
> > From Assumptions 3.5 and 3.6 it seems that the primary motivation is that the experts are themselves machines, making the motivating application in Example 3 incredibly problematic, especially from an ethics perspective: are we going to collect data from sepsis patients based on automated AI experts providing treatments by taking actions and observing trajectories? I don't think this is what the authors mean to suggest but providing this as a motivating application but then allowing unfettered querying of policies are not compatible. Please remove this example or provide a better example that is more compatible with the solution setup.
>
> We agree with the reviewer that Assumptions 3.5 and 3.6 do not allow the experts to be humans at the moment. We are planning to address this in future work where we aim to relax these assumptions. For instance, to relax Assumption 3.5, we believe that it might be possible to replace expert policies by their behavior-cloned versions (by training each on the given private data), with a slight loss in utility. We direct the reviewer to the response to Question 2 by reviewer CcoS for further discussion on assumptions.
>
> We would be happy to provide alternate motivating examples where the experts are machines. One example could be the case of financial portfolio management. Different fund managers employ proprietary and privacy-sensitive investment strategies that guide their trading decisions. In such a setting, treating the algorithms used by individual fund managers as expert policies, learning an expert-level private investment strategy (policy) using offline RL will be beneficial as a starting strategy for new investors and other parties that do not have specialized investment strategies.
>
> Another high-level scenario where we would be dealing with machine experts is whenever we need a student model to be learned from an ensemble of teacher models, in a privacy-preserving manner. In such a setting, these teacher models would form our expert policies. The PATE (Private Aggregation of Teacher Ensembles) method introduced in [8] also deals with a similar problem setting. We do not, however, assume access to unlabelled non-private data like in PATE. This makes our approach theoretically interesting.
>
> We hope that we were able to resolve the reviewer's concerns and would be happy to take more question if not so.
>
> **References:**
>
> [8] Papernot, Nicolas, et al. "Scalable private learning with pate." arXiv (2018).

---

> > ### Comment · Reviewer_rtSW · 2025-09-06
> > **Thank you for your detailed responses**
> >
> > I thank the authors for their detailed responses and clarifications. I clearly had some misunderstandings about the setup in the paper. That being said, I think I was not entirely negligent so I would encourage the authors to consider whether they can clarify in revision those places where I had some misunderstanding. Examples include:
> >
> > * The paper says clearly (on page 3) that "$r \colon S \times A \to \mathbb{R}$ is the (deterministic) reward function" so I am not sure why the authors now say "[w]e do not make the assumption that rewards are deterministic. " What am I supposed to believe?
> >
> > * Thank you for the clarification about the trajectory lengths. This is a case where the gap between how the algorithm is described versus how it is implemented could be made more clear. That is, what are **assumptions** versus **solution choices**. I think this will make things much clearer to future readers.
> >
> > Turning to the more substantive points:
> >
> > * If the intended application of **this** (and not future) work is to scenarios like the portfolio management example in the last section of the response, then this should be 100% clear and up front from the beginning of the article. I think that the gap between the setting here and the setting where there are real experts is nontrivial (and the authors seem to agree) so I think the title is a bit misleading: the only kinds of experts this works for are mechanical experts and the picture in Figure 1 (which shows the experts as people) is misleading. This is important because it affects what privacy *means* here: in finance the expert who is being protected is a company, not a person and thus privacy means some more like trade secrecy.
> >
> > * I think showing results like they have given above that make some high level points ("using the unstable set is useful") will help the experiments feel less like a "look, it works" and leave the reader with some more insight. This is more of a writing thing.
> >
> > * I agree that privacy amplication by subsampling is related but the argument/analysis has not been formalized. I think that a dedicated paragraph in the experiments discussing differences in description and implementation is needed. It's not enough to just appeal to an argument like "everyone else does it this way" + "the shuffle model will take care of the issues" without proof. As long as the manuscript is up front about the differences, the authors can the shuffle analysis to future work.

---

> ### Author Response · Authors · 2025-09-07
> **Thank you for your response!**
>
> We would like to thank the reviewer for their meaningful contribution to this discussion as well as their thoughtful comments and suggestions.
>
> > The paper says clearly (on page 3) that "$r \colon S \times A \to \mathbb{R}$ is the (deterministic) reward function" so I am not sure why the authors now say "[w]e do not make the assumption that rewards are deterministic. " What am I supposed to believe?
>
> We apologize for the confusion here. To make it clear, we DO NOT assume deterministic rewards in our setting. The line the reviewer points out (in page 3) is incorrect and we will correct it in the next revision of the paper. Thank you very much for bringing this to our attention.
>
> > Thank you for the clarification about the trajectory lengths. This is a case where the gap between how the algorithm is described versus how it is implemented could be made more clear. That is, what are assumptions versus solution choices. I think this will make things much clearer to future readers.
>
> We thank the reviewer for bringing this to our notice. We would be happy to update the description of Algorithm 2 (line 7 onwards) to make it clear that the algorithm works for variable length trajectories, which is how it is actually implemented in our experiments. We will also make it more precise in Section 5 that the analysis assumes same length trajectories for clarity, but extension is trivial by replacing $L$ with the length of the longest trajectory.
>
> > If the intended application of this (and not future) work is to scenarios like the portfolio management example in the last section of the response, then this should be 100% clear and up front from the beginning of the article. I think that the gap between the setting here and the setting where there are real experts is nontrivial (and the authors seem to agree) so I think the title is a bit misleading: the only kinds of experts this works for are mechanical experts and the picture in Figure 1 (which shows the experts as people) is misleading. This is important because it affects what privacy means here: in finance the expert who is being protected is a company, not a person and thus privacy means some more like trade secrecy.
>
> We would like to point out that in the Offline RL literature, expert policies refer to reasonably well-performing policies - this is the terminology we adopt here. However, we acknowledge the reviewer’s concerns that the use of the term “expert” may lead to a possible confusion with human experts, especially in settings where real human experts (like doctors etc.) are involved. In particular, we would be happy to update Figure 1 and denote experts using a non-human symbol. Further, we shall make this definition explicit in the introduction and add the examples discussed during the rebuttal to ensure that readers are clear about the settings in which our proposed method will be useful. We hope these changes will sufficiently address the reviewer’s concerns.
>
> > I think showing results like they have given above that make some high level points ("using the unstable set is useful") will help the experiments feel less like a "look, it works" and leave the reader with some more insight. This is more of a writing thing.
>
> We would be happy to add the results we discuss in the rebuttal to the manuscript, along with the relevant discussions, to add more depth to the takeaways.
>
> > I agree that privacy amplication by subsampling is related but the argument/analysis has not been formalized. I think that a dedicated paragraph in the experiments discussing differences in description and implementation is needed. It's not enough to just appeal to an argument like "everyone else does it this way" + "the shuffle model will take care of the issues" without proof. As long as the manuscript is up front about the differences, the authors can the shuffle analysis to future work.
>
> We agree with the reviewer that the gap between analysis and implementation should be discussed explicitly in the paper. We would be happy to add a discussion around this to the experiments section in the paper where we clearly describe the sampling differences between the pseudocode and the actual implementation (as stated in Response 4/5 above). We shall also add references to existing work in the “amplification-by-shuffling” vein of research.
>
>
> We would be happy to update the paper as per all these discussions during the revision phase.

---

### Comment · Action_Editor_tqT9 · 2025-08-12
**Please engage with authors**

Dear Reviewers, the discussion period has now begun. Please read the author responses and engage with the authors. Thank you!

---

### Decision · Action_Editor_tqT9 · 2025-09-23

**Recommendation:** Accept with minor revision

**Additional Comments:**

Reviewers had some concerns about the motivation of the work and its practical relevance. I recommend the authors make the changes that the reviewers suggested to improve clarity.

**Audience:**

Yes

**Audience Explanation:**

The reviewers agree that the setting is interesting and the proposed algorithm is not obvious. They particularly highlight that the modification to SVT may be of independent interest.

**Claims And Evidence:**

Yes

**Claims Explanation:**

Reviewers agree that the privacy proof is non-trivial and do not detect any correctness issues. The experimental evaluations show strong evidence that the algorithm outperforms DP-SGD.